# A searchable atlas of pathogen-sensitive lncRNA networks in human macrophages

Nils Schmerer [1], Harshavardhan Janga[1], Michelle Aillaud[1], Janina Hoffmann[1], Marina Aznaourova[1], Sarah Wende [1], Henrike Steding [1], Luke D. Halder[1], Michael Uhl [2,3], Fabian Boldt[1], Thorsten Stiewe[4,5,6], Andrea Nist[5], Lukas Jerrentrup[1], Andreas Kirschbaum[7], Clemens Ruppert [4,8,9], Oliver Rossbach[10], Evgenia Ntini [11], Annalisa Marsico [11,12], Chanil Valasarajan[8,13,14], Rolf Backofen [2,3], Uwe Linne [15], Soni S. Pullamsetti [4,6,8,13,14], Bernd Schmeck [1,4,6,16,17] & Leon N. Schulte [1,4] ✉

Long noncoding RNAs (lncRNA) are crucial yet underexplored regulators of human immunity. Here we develop GRADR, a method integrating gradient profiling with RNA-binding proteome analysis, to map the protein interactomes of all expressed RNAs in a single experiment to study mechanisms of lncRNA-mediated regulation of human primary macrophages. Applying GRADR alongside CRISPR-multiomics, we reveal a network of NFκB-dependent lncRNAs, including LINC01215, AC022816.1 and ROCKI, which modulate distinct aspects of macrophage immunity, particularly through interactions with mRNA-processing factors, such as hnRNP proteins. We further uncover the function of ROCKI in repressing the messenger of the anti-inflammatory GATA2 transcription factor, thus promoting macrophage activation. Lastly, all data are consolidated in the SMyLR web interface, a searchable reference catalog for exploring lncRNA functions and pathway-dependencies in immune cells. Our results thus not only highlight the important functions of lncRNAs in immune regulation, but also provide a rich resource for lncRNA studies.

The human genome is pervasively transcribed, producing a plethora of RNAs, most of which remain uncharacterized[1–3]. Among these, long non-coding RNAs (lncRNAs), arbitrarily defined as non-protein-coding RNAs longer than 200 nucleotides[4], have garnered increasing attention due to their critical roles in various cellular processes, including X chromosome inactivation and immune responses to infectious agents[5,6]. Despite the known importance of lncRNAs in the immune system, our current understanding of host-defense against pathogens predominantly rests on the functions of proteins. In the innate immune system, which constitutes the first line of defense against infections, various protein signaling pathways pivotal to pathogen recognition have been determined. For instance, macrophages, which are among

[1]Institute for Lung Research, Philipps University Marburg, 35043 Marburg, Germany. [2]Bioinformatics Group, Department of Computer Science, University of Freiburg, 79110 Freiburg, Germany. [3]Signalling Research Centre CIBSS, University of Freiburg, 79104 Freiburg, Germany. [4]German Center for Lung Research (DZL), 35392 Giessen, Germany. [5]Genomics Core Facility, Institute of Molecular Oncology, University of Marburg, 35043 Marburg, Germany. [6]Institute for Lung Health (ILH), Justus-Liebig University, Giessen, Germany. [7]Department of Visceral, Thoracic and Vascular Surgery, University Hospital Giessen and Marburg (UKGM), Marburg, Germany. [8]Universities of Giessen and Marburg Lung Center (UGMLC), Giessen 35392, Germany. [9]UGMLC Giessen Biobank and european IPF registry (eurIPFreg), Giessen 35392, Germany. [10]Institute for Biochemistry, FB08, Justus Liebig University Giessen, 35392 Giessen, Germany. [11]Max Planck Institute for Molecular Genetics, 14195 Berlin, Germany. [12]Institute for Computational Biology, Helmholtz Center, 85764 München, Germany. [13]Max Planck Institute for Heart and Lung Research, Bad Nauheim, Germany. [14]Excellence Cluster Cardio-Pulmonary Institute (CPI), Justus-Liebig University, Giessen, Germany. [15]Mass spectrometry facility of the Department of Chemistry, Philipps University, Marburg, Germany. [16]Department of Medicine, Pulmonary and Critical Care Medicine, University Hospital Giessen and Marburg, Philipps University Marburg, Marburg, Germany. [17]German Centre for Infectious Disease Research (DZIF), SYNMIKRO Centre for Synthetic Microbiology, Philipps University Marburg, Marburg, Germany. ✉e-mail: leon.schulte@uni-marburg.de

the first cells to sense an infection, use pattern recognition receptors (PRRs) such as Toll-like receptor TLR4, which detects lipopolysaccharide (LPS), a major component of gram-negative bacterial cell walls. TLR4 activation triggers the MyD88-NFκB signaling pathway, essential for the production of pro-inflammatory immune mediators like IL-1β and IL-8[7]. In parallel, TLR4 triggers the TRIF-IRF3 signaling pathway, which promotes the production of type I interferons (IFN-I), conferring e.g., protection against intracellular infections[7]. A multitude of additional PRRs contribute to pathogen detection by the immune system.

Recent studies have highlighted significant involvement of specific lncRNAs, such as MaIL1 and LUCAT1, in these innate immune-signaling pathways. MaIL1 enhances TRIF-IRF3 signaling by binding and stabilizing the protein Optineurin (OPTN), leading to IFN-I production[8]. LUCAT1 acts as a negative feedback regulator of this pathway, likely via hnRNP and STAT1 protein interactions[9–12]. Other lncRNAs such as GAPLINC, PACERR, and NEAT1 are known to contribute to antimicrobial defense by regulating the pro-inflammatory immune response[13–16]. Additional lncRNAs have been implicated in human innate immunity[6,17]. However, despite these discoveries, the vast majority of lncRNAs expressed in human immune cells remains uncharacterized[8], likely due to the challenging discovery of lncRNA-interactions with other biomolecules, required for deciphering their modes of action. While methods such as RAP-MS or SHIFTR exist to explore individual lncRNA interactions with proteins[18,19], they typically lack high-throughput capability. Recently, we used Grad-seq to categorize immune-activated macrophage lncRNAs into various groups based on their co-sedimentation with major protein machineries[8], yet this approach falls short of providing detailed interaction data for individual lncRNAs, needed for deeper mechanistic investigations. These limitations coupled with a lack of comprehensive reference data detailing the molecular pathways controlling lncRNA expression, impede advances in our understanding of the timing and functioning of lncRNAs in defense against infections and immune-pathologies in humans.

In this work, we map the lncRNA landscape of primary human macrophages and introduce GRADR, a method that globally predicts RNA-protein interactions based on gradient co-sedimentation. By coupling GRADR with CRISPR multiomics, we reveal regulatory functions for multiple previously uncharacterized lncRNAs in innate immunity. Our findings are consolidated in a web interface, the Searchable Myeloid LncRNA Registry (SMyLR; rna-lab.org/smylr). We believe that this resource will significantly advance the characterization of RNA-mechanisms in human immunity. Our results may also guide future efforts to target lncRNA networks in immune-related diseases.

## Results

### Human lung immunity involves numerous largely uncharacterized lncRNAs

Long non-coding RNAs (lncRNAs) are emerging as pivotal regulators in eukaryotic biology, yet their functions in human immunity are only beginning to be defined. In this study, we present a complex multiomics framework for large-scale lncRNA characterization in human cells, revealing mechanisms of RNA-dependent immune-regulation. We further introduce the Searchable Myeloid lncRNA Registry, SMyLR (rna-lab.org/smylr), an open-access resource detailing lncRNA pathway dependencies, subcellular localization and interactions with proteins during primary macrophage immune activation (Fig. 1A, B). We specifically examined key cell types of the delicate alveolar barrier, a primary site of pathogen-attack, to define core lncRNA networks relevant to human host defense.

Our study utilized human type II alveolar epithelial cells (AECII) derived from resected lung tissue (Supplementary Fig. 1A) and alveolar macrophages (aMΦ) from healthy volunteers via bronchoalveolar lavage. We also differentiated CD14+ monocytes into macrophages using M-CSF or GM-CSF (M-MΦ or G-MΦ) to simulate infection-driven monocyte recruitment and differentiation. While optimizing conditions for RNA-seq profiling of mRNAs and lncRNAs, we observed distinct immune responses of these cells to bacterial stimuli. Both AECII and macrophages responded to pathogenic *Legionella pneumophila* and non-pathogenic *Escherichia coli* (Supplementary Fig. 1B), yet through different pathways. Our findings show that macrophages display a robust response to several Toll-like receptor ligands, notably TLR4, TLR5, and TLR2, while AECII cells only responded significantly to TLR5 stimulation (Supplementary Fig. 1B and Fig. 1C, D). As expected, RNA-seq revealed the three investigated types of macrophages (aMΦ, M-MΦ and G-MΦ) to display a more similar mRNA response to TLR-stimulation than AECII cells (Fig. 1C, D, Supplementary Data 1). Yet, a limited set of key NFκB-driven immune genes were activated both in AECII and aMΦ (e.g. CSF2, IL6, IL23A), underscoring the division of labor between epithelial cells and macrophages during human lung immunity (Supplementary Fig. 1C, D).

Comparative analysis suggested that M-MΦ and G-MΦ recapitulate different aspects of the aMΦ response to LPS. For example, expression of the cytokine IL10 was induced in both aMΦ and M-MΦ, while the coagulation factor SERPINB2 was induced in aMΦ and G-MΦ (Supplementary Fig. 1E-F). Despite the recorded differences, all three macrophage types shared the induction of several key immune response factors, such as CCL4 or IL6 (Supplementary Fig. 1E, F). Examining lncRNA regulation across the different macrophage types, we found that most of the 39 lncRNAs upregulated ≥ 2-fold in aMΦ following LPS stimulation (in both RNA-seq replicates) were also induced in M-MΦ and G-MΦ (Supplementary Fig. 2A). Thus, although considerable differences exist, all three macrophage types exhibit a common immune-response signature at both the mRNA and lncRNA levels. For this study, we selected G-MΦ as our primary in vitro model, due to its scalable availability and the essential role of GM-CSF in alveolar macrophage replenishment in vivo[20]. Among the lncRNAs regulated in all analyzed cell types (Supplementary Fig. 2A), 24 were consistently up-regulated ≥ 2-fold in both RNA-seq replicates in aMΦ and G-MΦ following LPS stimulation. This group included known immune-regulatory lncRNAs (e.g. MaIL1 and PACERR), as well as several previously poorly characterized lncRNAs, such as ROCKI, AC010980, LINC00158, LINC01215, and AC022816.1 (Fig. 1E, F, Supplementary Fig. 2A). Re-analysis of published RNA-seq datasets[21–23] further verified the up-regulation of these lncRNAs (≥2-fold on average) upon macrophage and monocyte stimulation with LPS (Supplementary Fig. 2B), with varying levels of significance (Supplementary Fig. 2C).

To validate the engagement of these five lncRNAs under conditions that better mimic the native immune-contexts, we conducted 4-and 8-hour stimulations (both LPS and flagellin) of human precision-cut lung tissue slices (PCLS) which preserve native tissue architecture and inter-cell-type communication (Fig. 1G). These experiments confirmed the up-regulation of the selected lncRNAs following either TLR4 (LPS) or TLR5 (flagellin) stimulation, with statistically significant induction observed under at least one condition in all cases except for LINC00158 (Fig. 1H). Furthermore, we quantified the expression of these lncRNAs in bronchoalveolar lavage (BAL)-derived cells from 24 patients undergoing BAL for pulmonary disease assessment. Our recent work demonstrated that IFNB1 serves as a sensitive marker of pulmonary immune activation in BAL studies[8]. Notably, all 5 lncRNAs - but not the control transcripts U6 snRNA or RPS18 mRNA - showed significant linear correlation with IFNB1 expression in the patient cohort, further supporting their involvement in pulmonary immunity (Fig. 1I, Supplementary Fig. 2D and E).

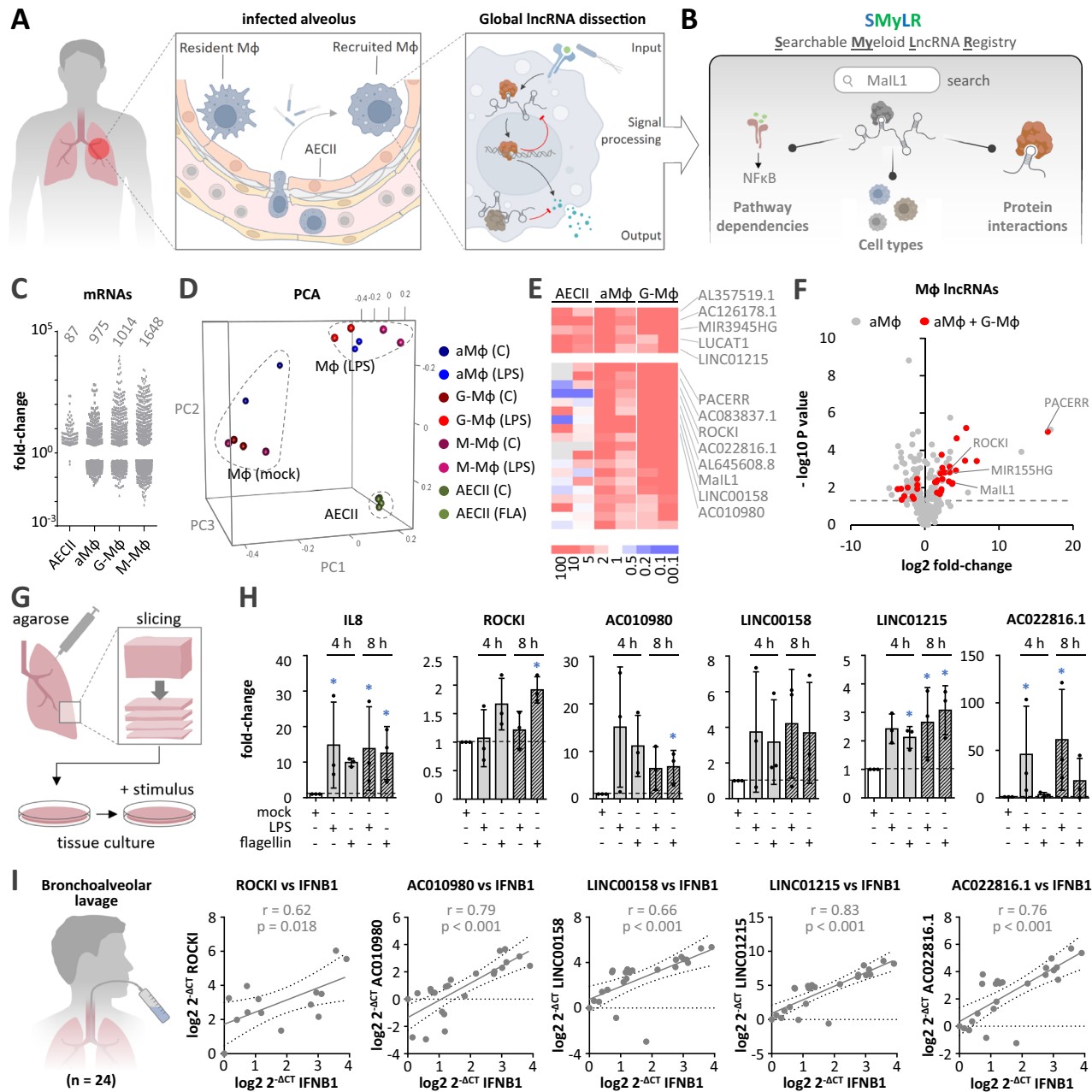

**Fig. 1 | Mapping of lncRNAs implicated in human lung innate immunity.**
**A** Illustration of key alveolar cell types, highlighting macrophage lncRNA networks. **B** Overview of the SMyLR database, detailing lncRNA pathway dependencies, cell-type specificity, and protein interactions. **C** Number and fold changes (≥2-fold up or down) of mRNAs in specified cell types following 4 h of flagellin (FLA) (AECII) or LPS (aMΦ, G-MΦ, M-MΦ) stimulation; data averaged from two RNA-seq replicates. **D** 3D PCA analysis of RNA-seq samples from (**C**). **E** Heatmap of all lncRNAs up-regulated ≥ 2-fold in both RNA-seq replicates in aMΦ or G-MΦ (data from **C**, **D** AECII data included for comparison). Top section highlights lncRNAs also upregulated in AECII ≥ 2-fold. **F** Volcano plot displaying lncRNA changes (4 h LPS vs control treatment) in aMΦ, highlighting those also regulated in G-MΦ (data from C-E). Two-tailed Student's t-test *p*-values are shown. **G** Illustration of PCLS preparation and culture. **H** RT-qPCR analysis of IL8 and lncRNA regulation in PCLS post stimulation (results relative to RPS18 and 4 h mock). Three independent experiments; mean values +/- SD. Asterisks denote statistical significance (One-way ANOVA test, *p* ≤ 0.05). Exact *p*-values: IL8: 0.017 (4 h LPS), 0.024 (8 h LPS), 0.039 (8 h FLA); ROCKI: 0.034 (8 h FLA); AC010980: 0.042 (8 h FLA); LINC01215: 0.045 (4 h FLA), 0.048 (8 h LPS), 0.017 (8 h FLA); AC022816.1: 0.025 (4 h LPS), 0.011 (8 h LPS). **I** Left: Illustration of BAL procedure. Right: Linear regression plots with 95% confidence intervals (dashed lines) and Pearson correlation statistics (two-tailed test), comparing the RT-qPCR-determined levels of the indicated lncRNAs with IFNB1 (log2 $2^{-\Delta CT}$ values) in bronchoalveolar lavage cell pellets. Samples size variations in individual plots (n ≤ 24) reflect cases where the respective lncRNA was below the detection limit.

Together, these results underscore the engagement of various macrophage lncRNAs in human anti-pathogen responses. To promote further research into their diverse roles we systematically dissected their pathway-dependencies, conservation, functions and protein-interactions (Fig. 2A).

## LncRNAs are embedded in different pathways and phases of the immune response

To elucidate the pathway dependencies of the identified immune-associated lncRNAs, we exposed G-MΦ to various immune agonists and pathway inhibitors, followed by RNA-seq. Furthermore, we tracked

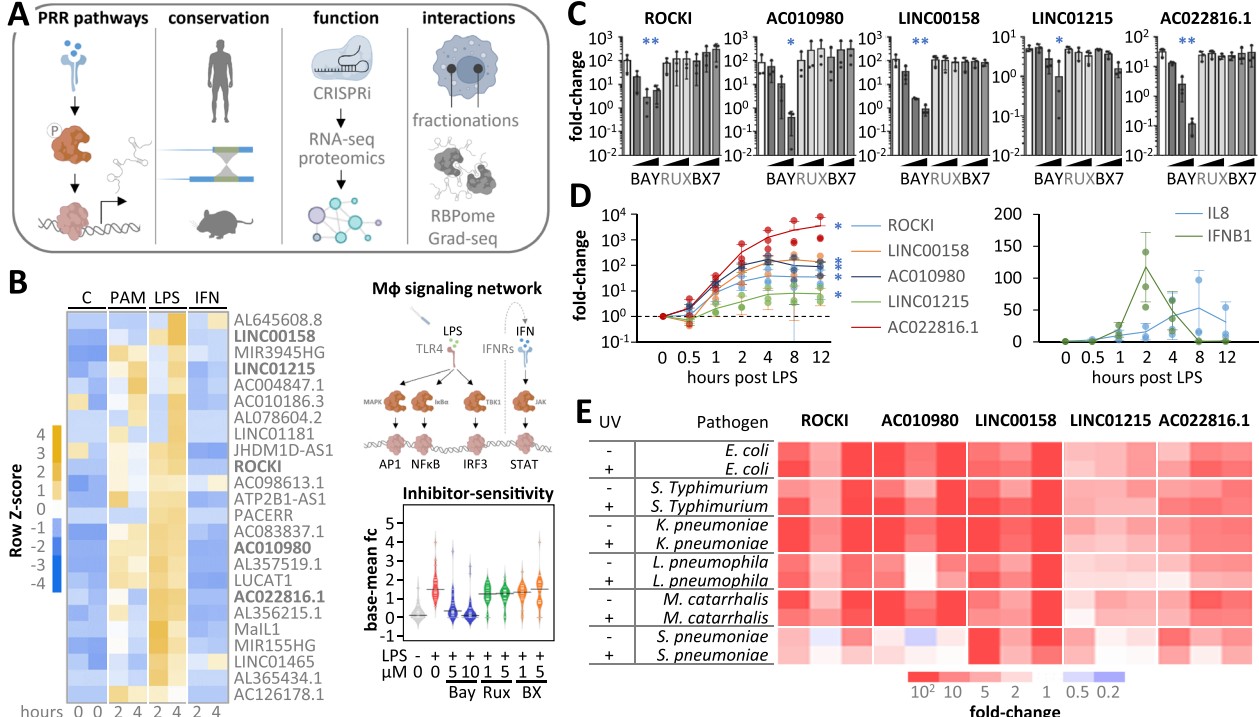

**Fig. 2 | Pathway dependency of macrophage lncRNAs. A** Illustration of attributes analyzed to characterize immune-responsive lncRNAs in macrophages. **B** Left: Heatmap (row Z-scores) showing lncRNA expression changes in G-MΦ following 2 h and 4 h stimulation with Pam3csk4 (PAM), LPS, or interferon-α (IFN), based on averaged RPKMs from two independent RNA-seq datasets. Right: Top: Illustration of pathways relevant for LPS-dependent lncRNA regulation. Bottom: RNA-seq analysis of the dependence of lncRNAs shown in the heatmap (left) on LPS-induced pathways. Bay = BAY 11-7082 (NFκB inhibitor); Rux = Ruxolitinib (STAT-inhibitor); BX = BX795 (TBK1-IRF3 inhibitor). Fold-changes relative to base-mean, averaged from two independent RNA-seq experiments. **C** RT-qPCR analysis of lncRNA responses to LPS (4 h) plus either DMSO or inhibitors used in B) at three increasing concentrations; results from three independent experiments; mean values +/- SD.

Exact *p*-values (in the same order as the asterisks): ROCKI: 0.001, 0.013; AC010980: <0.001; LINC00158: <0.001, <0.001; LINC01215: 0.001; AC022816.1: <0.001, <0.001. **D** Time-series RT-qPCR analysis of IL8, IFNB1, and lncRNA expression in G-MΦ treated with LPS; data from three independent experiments, showing mean values +/- SD. Exact *p*-values: ROCKI: <0.001; LINC00158: <0.001; AC010980: <0.001; LINC01215: <0.001; AC022816.1: 0.004. **E** Heatmap of lncRNA changes in G-MΦ exposed to various live or UV-inactivated bacterial pathogens for 4 h; three independent experiments; RT-qPCR. **C**, **D**: One-way ANOVA tests were conducted. **C**: all inhibitor treatments were compared to the LPS-treated control. **D** (left panel): *P*-values were calculated for the linear trends of column averages from left to right. Statistical significance ($p \leq 0.05$) is indicated by asterisks.

lncRNA temporal regulation by performing a time-course LPS stimulation experiment coupled to RNA-seq at intervals from 0.5 to 12 hours. Most lncRNAs found to be up-regulated by LPS in aMΦ and G-MΦ (Fig. 1E, Supplementary Fig. 2A), also responded to TLR2 agonist Pam3CSK4 but not to type I IFN (IFNα) (Fig. 2B, Supplementary Fig. 3A–D, Supplementary Data 2). Inhibition of NFκB-dependent gene expression with BAY-11-7082 globally reduced up-regulation of these lncRNAs. In contrast, inhibition of STAT and IRF signaling with Ruxolitinib and BX795 respectively, did not affect global lncRNA induction in macrophages in response to LPS (Fig. 2B, Supplementary Fig. 3E, Supplementary Data 3). These findings suggest that most immune-inducible lncRNAs are regulated in an NFκB-dependent manner. RT-qPCR validations for the five lncRNAs of primary interest (see above) confirmed a consistent NFκB dependency, with only LINC01215 showing additional sensitivity to the TBK1-IRF3 pathway in BX795 inhibitor treatments (Fig. 2C and Supplementary Fig. 3E). Temporally, these lncRNAs followed regulation kinetics similar to the classic immune response gene *IL8 (CXCL8)* in G-MΦ (Fig. 2D, Supplementary Fig. 3C and F). However, other lncRNAs, such as MaIL1 and PACERR, exhibited earlier expression peaks (e.g. at 1 hour post-LPS challenge) (Supplementary Fig. 3F), suggesting that lncRNAs play distinct roles at different stages of the macrophage immune response. Notably, similar lncRNA regulation kinetics were observed in M-MΦ, with the exception of LINC01215, which was not regulated in this macrophage type (Supplementary Fig. 3G). RT-qPCR experiments further confirmed up-

regulation of the five lncRNAs in focus in G-MΦ in response to multiple live or UV-inactivated bacterial pathogens, including *S. Typhimurium*, *K. pneumoniae*, *M. catarrhalis*, and *S. pneumoniae* (Fig. 2E), underscoring their involvement in host defense. Comparative genomic analysis revealed that while the five lncRNAs in focus, as well as macrophage LPS-inducible lncRNAs in general (lncRNAs from Fig. 1E), are conserved among primates, their conservation is limited in mice. However, better conservation was observed in species with longer generation times (Fig. 3A).

Taken together, these findings suggest that the lncRNA responses depicted here reflect a common core program with minor variations across macrophage types. The limited evolutionary conservation highlights a potential challenge in using murine models to study the in vivo functions of many human macrophage lncRNAs, including those central to this study.

## CRISPR-multiomics uncovers a broad lncRNA network regulating innate immunity

To estimate the regulatory potential and redundancy among various immune-responsive lncRNAs, we conducted loss-of-function studies focusing on the lncRNAs selected for validation above (ROCKI, AC010980, LINC00158, LINC01215, and AC022816.1). Using CRISPR interference (CRISPRi), we silenced each lncRNA using ≥ 2 independent guideRNA designs in THP1 macrophages, followed by LPS stimulation, RT-qPCR and RNA-seq analysis in triplicates. All five lncRNAs

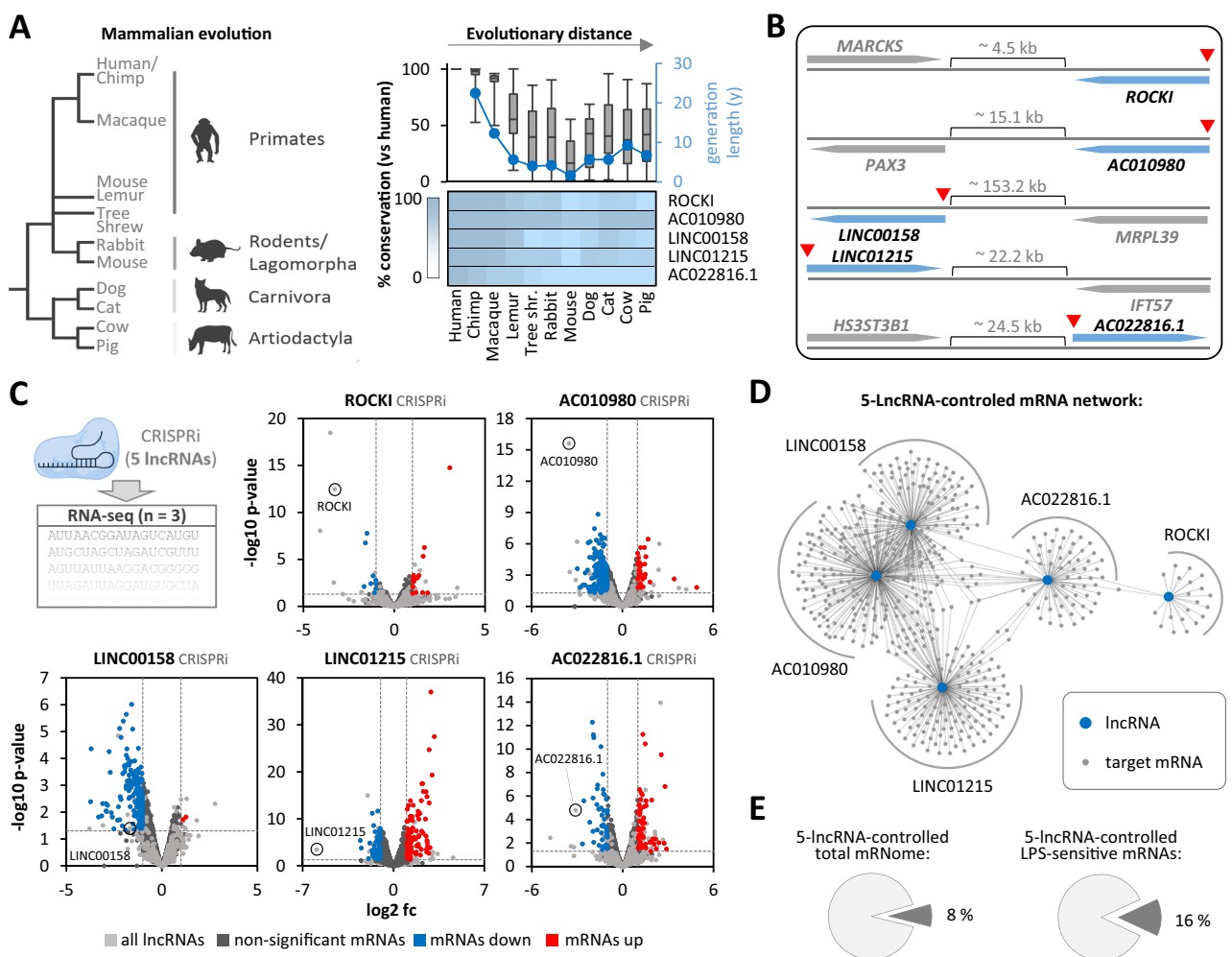

**Fig. 3 | Conservation and functional impact of macrophage lncRNAs. A** Left: Evolutionary tree of selected mammalian species. Right: Percent base conservation of human LPS-inducible macrophage lncRNA sequences across the genomes of indicated species, sorted by evolutionary distance; top panel includes all 24 lncRNAs from Fig. 1E and data are shown as box plots (median and 75th-25th percentile interquartile range, whiskers indicate minimum and maximum data values), with species generation time overlayed; bottom panel focuses on specific lncRNAs. **B** Illustration of genomic locations of specified lncRNAs relative to nearest neighboring genes (distance in kilobases [kb] provided). Red triangles indicate transcriptional start site positions targeted by CRISPRi for lncRNA silencing.

**C** Volcano plots from CRISPRi-based lncRNA loss-of-function experiments in THP1 cells stimulated with LPS for 8 h. Fold-changes (fc) compare lncRNA-knockdown cells to empty vector control cells. Results from three independent experiments. Two-tailed Student's t-test *p*-values are shown. **D** Cytoscape network of lncRNAs (blue) and mRNAs from panel C, regulated upon lncRNA silencing (≥2-fold up or down, *p* ≤ 0.05, two-tailed Student's t-test). **E** Pie charts showing proportions of all expressed mRNAs or LPS-responsive mRNAs (≥2-fold up or down, *p* ≤ 0.05, two-tailed Student's t-test) affected by silencing of one or more lncRNAs. **C-E:** 8 h LPS-stimulated THP1 cells and three independent replicates.

were located several kilobases away from neighboring genes, suggesting that directing the CRISPRi machinery to the transcriptional start site regions of these lncRNAs is unlikely to directly affect the expression of adjacent genes (Fig. 3B). RT-qPCR and RNA-seq analysis confirmed successful lncRNA silencing (Supplementary Fig. 4A, Fig. 3C, Supplementary Data 4) and revealed a broad mRNA network significantly influenced by the five lncRNAs in immune-challenged cells (Fig. 3C, D, Supplementary Fig. 4B). Notably, substantial overlap in mRNA regulation was observed between LINC00158 and AC010980, suggesting partial redundancy between these two lncRNAs, whereas AC022816.1, LINC01215, and ROCKI influenced more distinct mRNA sets (Fig. 3D). Overall, ~8% of all expressed mRNAs (RPKM ≥ 0.5) and ~16% of all LPS-responsive mRNAs (fold-change ≥ 2) were under regulatory influence by the 5 lncRNAs (≥2-fold change and *p* ≤ 0.05 upon loss of ≥ 1 lncRNA) (Fig. 3E).

To correlate mRNA regulation with protein abundance, we supplemented our lncRNA loss-of-function RNA-seq data with quantitative whole-proteome analysis. Significant correlations between regulated

mRNAs and their corresponding proteins were noted for LINC01215 and AC022816.1 ($p < 10^{-4}$), with ROCKI approaching significance ($p = 0.06$) (Fig. 4A, B, Supplementary Data 5). LINC00158 and AC010980 showed a less pronounced impact at the protein level (Fig. 4A, B). Notably, when restricting the analysis to factors regulated both at the RNA and the protein level upon lncRNA-loss, the five lncRNAs impacted the expression of distinct sets of genes, with LINC01215 and AC022816.1 mostly impinging on immune-related pathways and LINC00158 and AC010980 on RNA-processing and metabolism-related pathways (Fig. 4B, C, Supplementary Fig. 5A-C). When limiting the analysis to LPS-responsive genes, ROCKI, LINC01215, and AC022816.1 were found to significantly influence the LPS response at the mRNA level, with LINC01215 also exerting significant effects at the protein level (Fig. 4D). Specifically, genes that were down-regulated upon ROCKI silencing were up-regulated in response to LPS, while genes up-regulated upon ROCKI silencing exhibited the opposite trend. In contrast, genes regulated by LINC01215 or AC022816.1 showed the opposite behavior (Fig. 4D), suggesting that

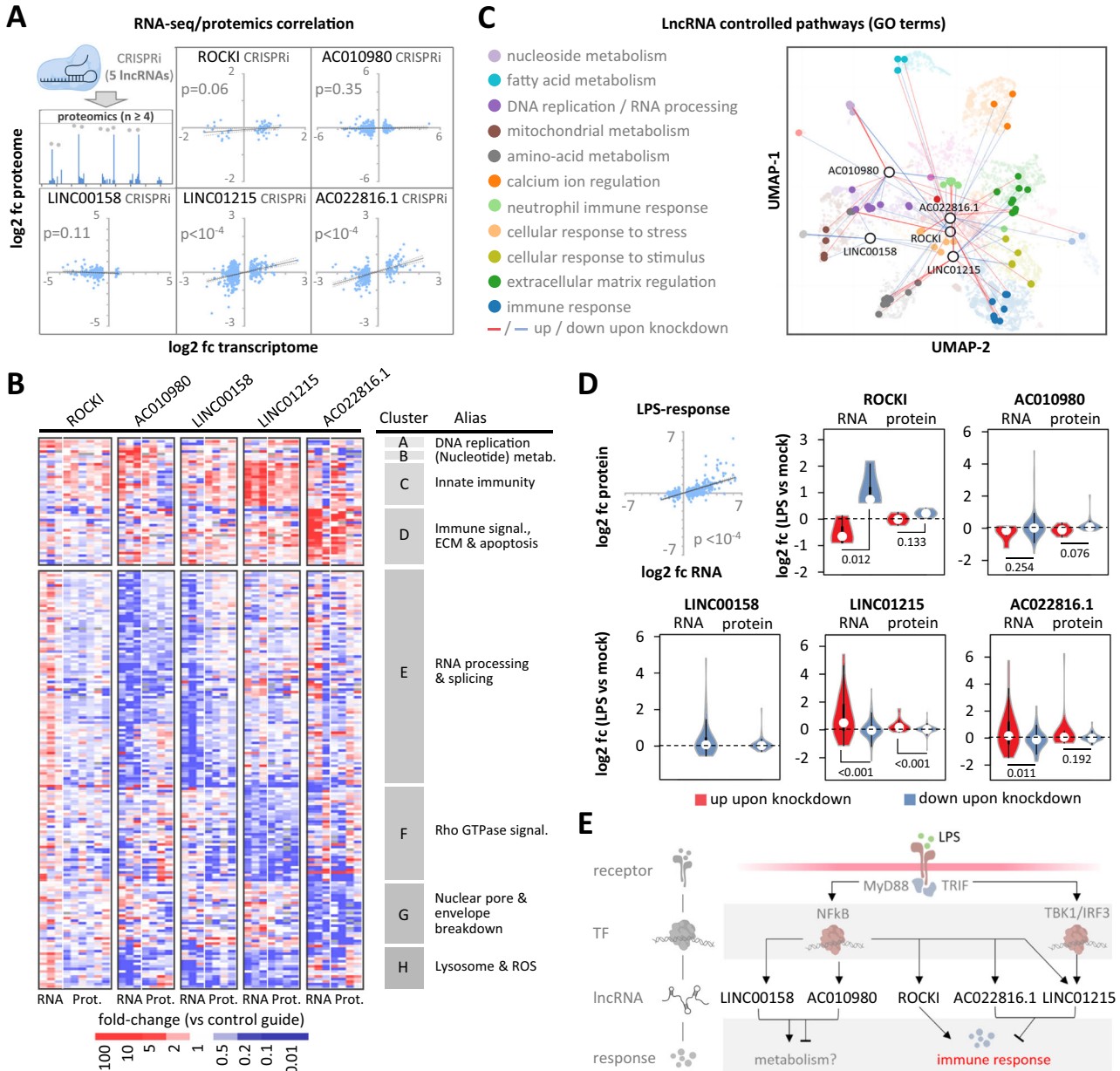

**Fig. 4 | Comparative analysis of lncRNA loss-of-function effects at RNA and protein levels. A** Pearson correlation analysis (*p*-values shown, two-tailed test) comparing RNA-seq-determined significant mRNA fold-changes (from Fig. 3) and corresponding protein fold-changes. **B** Clustered heatmap showing aggregated factors regulated at both RNA and protein levels (≥2-fold up or down, *p* ≤ 0.05) upon lncRNA silencing. Clusters (A-H) and their primary Reactome pathways (Supplementary Fig. 5A) are indicated. **C** ENRICHR-based UMAP visualization of the "Biological Process" GO-term network, highlighting GO-terms associated with gene products that are significantly regulated at the RNA and protein levels (≥2-fold upregulation [red lines] or downregulation [blue lines], *p* ≤ 0.05, two-tailed

Student's t-test) upon silencing of the depicted lncRNAs. **D** Top left: correlation of mRNA (≥2-fold up or down, *p* ≤ 0.05, two-tailed Student's t-test) and corresponding protein fold-changes (RNA-seq and proteomics, ≥ 3 independent experiments) in response to LPS (8 h, THP1 cells). Pearson correlation *p*-value (two-tailed test) is shown. Violin plots: fold-changes at the RNA- and protein-level in response to LPS of factors regulated upon lncRNA silencing (≥2-fold up [red] or down [blue], *p* ≤ 0.05 in data from **A**, **B**). Two-tailed Student's t-test *p*-values are shown. **E** Schematic summary of hypothesized functions for the five studied lncRNAs in macrophages. All panels: 8 h LPS-stimulated THP1 cells and ≥ three independent replicates.

ROCKI acts to enhance, whereas LINC01215 and AC022816.1 function to suppress the LPS-response.

Collectively, our comprehensive, comparative lncRNA loss-of-function studies highlight LINC01215 and AC022816.1 as negative regulators (suppressors) of macrophage immunity, while suggesting that ROCKI serves as a positive regulator (enhancer) (Fig. 4E). However, unlike LINC01215, the regulatory effects observed at the RNA level for ROCKI and AC022816.1 appear to translate to the protein level only to a limited extend. During the preparation of our manuscript, ROCKI was independently identified as a positive regulator of

macrophage immunity, corroborating our findings[24]. To explore the potential modes of action of macrophage lncRNAs, we determined their cooperation with protein partners.

### GRADR charts the global lncRNA-protein interaction landscape in macrophages

A primary challenge in lncRNA research is identifying protein interaction partners, a labor-intensive process reliant on individual affinity purification of each lncRNA. To accelerate the characterization of lncRNAs compiled in our catalog, we developed a streamlined approach termed

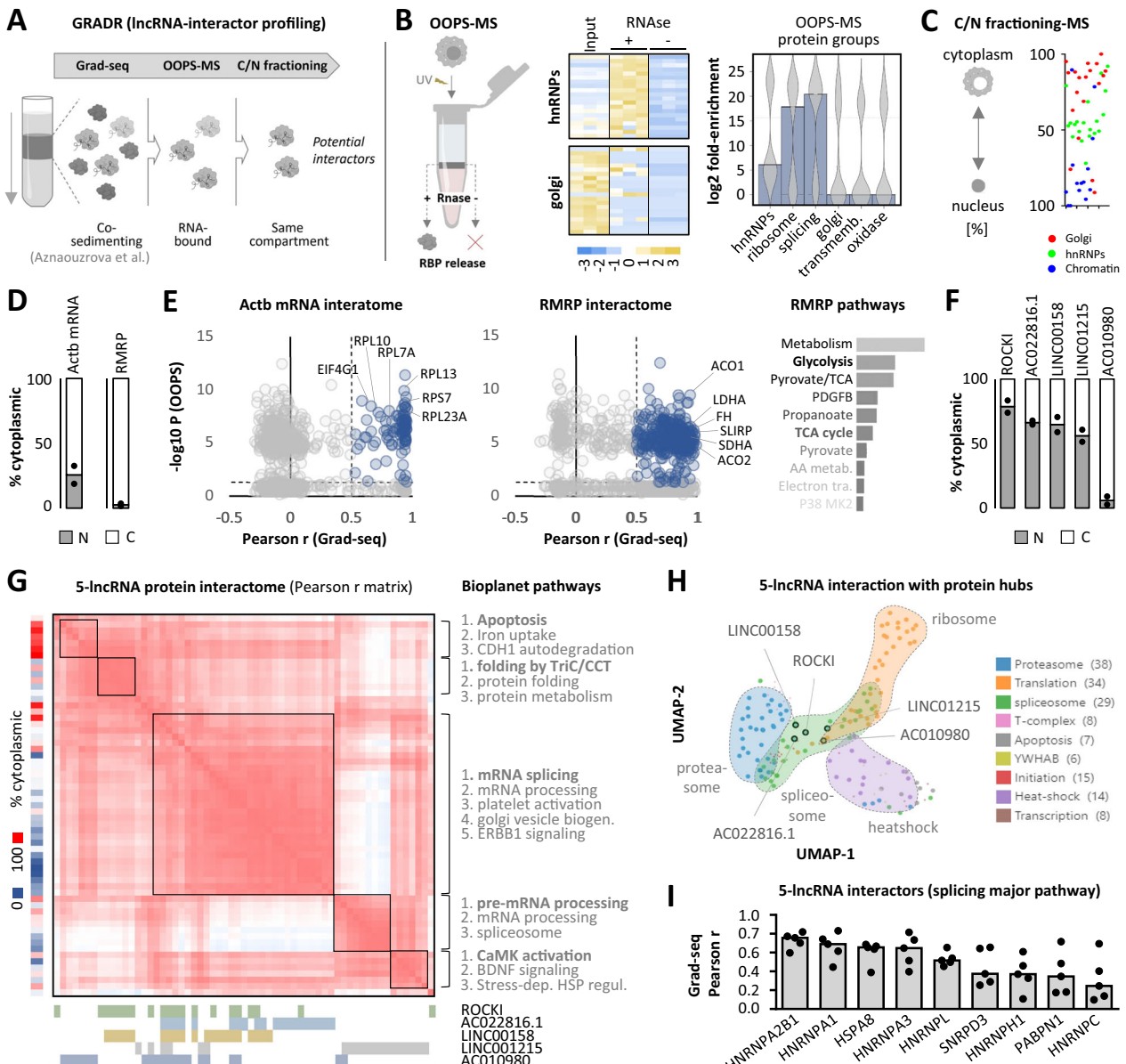

**Fig. 5 | Development of GRADR for predicting lncRNA-protein interactions in macrophages. A** Illustration of the GRADR methodology. **B** Z-score heatmap and violin plot (overlayed averages as bars) displaying differential enrichment of RNA-binding and non-RNA-binding protein groups in OOPS-MS eluates (violin plot: averages from three independent experiments). **C** Illustration of the distribution of representative protein classes within cytoplasmic and nuclear fractions; data averaged from three independent proteomics experiments. **D** Percentage of *ACTB* mRNA and RMRP lncRNA across cytoplasmic (C) and nuclear (N) fractions (RNA-seq, averages and individual data points from two independent experiments). **E** Panels 1-2: GRADR-derived protein interactome for *ACTB* mRNA and RMRP lncRNA, plotting co-sedimentation Pearson r values against -log10 OOPS-MS RNA-binding p-values (two-tailed Student's t-test). Proteins with r ≥ 0.5 and p ≤ 0.05 highlighted in blue. Panel 3: Reactome pathway analysis for proteins associated with RMRP (blue in panel 2). **F** Same as **D**), but for the indicated lncRNAs. **G** Grad-seq Pearson correlation matrix for proteins identified by GRADR (r ≥ 0.5 and p ≤ 0.05, two-tailed Student's t-test) as potential interactors of the five lncRNAs shown in F (Pearson r color coded from red to white [1–0]). % cytoplasmic localization from **C** indicated to the left, protein-lncRNA interactions are noted below. Relevant clusters were subjected to Bioplanet pathway analysis (shown to the right). **H** UMAP plot displaying the glycerol gradient distribution (Grad-seq) of the five lncRNAs (highlighted as circles with black outlines) and proteins belonging to several cellular machineries. Relevant machineries highlighted for context. **I** Grad-seq Pearson r values determined for the indicated splicing-associated proteins (from **G**) and the five lncRNAs. **All panels:** 4 h LPS-stimulated G-MΦ.

GRADR (<u>G</u>radient-based <u>R</u>NA interactor profiling), predicting the interactomes of all expressed lncRNAs in a single experiment. GRADR integrates several techniques (Fig. 5A): initially, Grad-seq separates cellular RNA-protein complexes by size on a linear glycerol gradient, followed by fractionation. Subsequent RNA-seq and proteomics analyses reveals the global co-sedimentation landscape, providing a comprehensive view of possible RNA and protein interactions. Additionally, OOPS-MS (Orthogonal Organic Phase Separation-Mass Spectrometry) refines this data by pinpointing those co-sedimenting proteins that have RNA binding

capacity and thus are likely interactors. Parallel cytoplasm/nucleus fractionations coupled to RNA-seq and proteomics focus the analysis on proteins and RNAs present in the same subcellular compartment. This approach narrows the protein interactor space for each expressed RNA to a few dozen candidates, as demonstrated below.

For the initial step of GRADR, we revisited our previously published Grad-seq co-sedimentome data from LPS-treated G-MΦ[8]. We complemented this dataset with OOPS-MS analysis using RNase-based RBP elution, which delineated the RNA-binding proteome in G-MΦ.

The results successfully discriminated between known RNA-binding and non-RNA-binding protein classes as positive and negative controls, respectively (Fig. 5B, Supplementary Fig. 6A). Further, subcellular fractionation combined with proteomics accurately mapped proteins to their respective cellular compartments, e.g., identifying chromatin factors in the nucleus and Golgi proteins in the cytoplasm (Fig. 5C, Supplementary Fig. 6B). To validate the efficacy of GRADR, we examined its ability to pinpoint proteins known to co-localize with cytoplasmic *ACTB* mRNA and the mitochondrial RMRP lncRNA (Fig. 5D, Supplementary Fig. 6C, Supplementary Data 6–8). GRADR accurately predicted interactions between *ACTB* mRNA and ribosomal proteins, and between RMRP and mitochondrial proteins involved in pathways such as glycolysis and the TCA cycle (Fig. 5E). To further benchmark GRADR against known interactomes of immune-regulatory lncRNAs in macrophages, we compared its predictions for the interferon-regulatory lncRNA LUCAT1 (Supplementary Fig. 6C and D) with ChIRP-MS affinity purifications conducted both in this study and previously published by others[10] (Fig. 6E–G, Supplementary Data 9). Our own ChIRP-MS data independently confirmed the previously reported association of LUCAT1 with hnRNP proteins and generally with splicing-related factors (Supplementary Fig. 6E and F)[10]. Consistently, GRADR also predicted interactions between LUCAT1, hnRNPs and splicing factors (Supplementary Fig. 6D and F), several of which had previously been identified as LUCAT1 interactors by Vierbuchen et al.[10]. (Supplementary Fig. 6G).

Having established the capability of GRADR to correctly predict known RNA-protein interactions, we applied it to identify proteins and protein machineries associated with the five immune-related lncRNAs in the focus of this study. Initial mapping of RNA subcellular localization via RNA-seq showed ROCKI, AC022816.1, LINC00158, and LINC01215 to be primarily nuclear, while AC010980 was found mainly in the cytoplasm (Fig. 5F). GRADR identified numerous potential interactors for these lncRNAs, notably chaperones and splicing factors (Fig. 5G–I, Supplementary Fig. 6H, Fig. 6A). Grad-seq further revealed a substantial cluster of co-sedimenting proteins among the GRADR-predicted interactors of these five lncRNAs, predominantly associated with splicing (Fig. 5G). UMAP analysis of the Grad-seq sedimentation patterns of the five lncRNAs and proteins belonging to various intracellular machineries supported the association of these lncRNAs with splicing factors (Fig. 5H). Within the Bioplanet splicing pathway, GRADR identified nine proteins, primarily from the hnRNP family, as potential core-interactors for the five lncRNAs (Fig. 5I). These findings are consistent with the established function of hnRNPs in lncRNA-mediated immunoregulation[10,25].

## LncRNA ROCKI cooperates with hnRNPs to remove a GATA2 dependent brake on immunity

In our CRISPR loss-of-function analyses, the lncRNA ROCKI emerged as a potential positive regulator of the macrophage immune response (Fig. 4). To further substantiate this role and its GRADR-predicted interaction with hnRNP proteins, we investigated the influence of ROCKI on macrophages and explored its mechanisms of action. Although ROCKI did not significantly alter the overall LPS response at the protein level (Fig. 4), detailed analysis revealed the down-regulation of key pro-inflammatory mediators upon ROCKI silencing, including IL1b, CCL20, IL8, and PTGS2 (Supplementary Fig. 7A). This regulation was confirmed through independent IL1β ELISA assays (Supplementary Fig. 7B), further suggesting that ROCKI acts as an NFκB-inducible feed-forward regulator of the inflammatory response (Supplementary Fig. 7C).

To elucidate its mechanism, we focused on the GRADR-predicted interaction of ROCKI with hnRNP proteins (Fig. 6A). To test the validity of the GRADR predictions, we employed ChIRP-MS RNA-antisense purification to determine ROCKI-bound proteins. ChIRP-MS validated the primary association of ROCKI with several hnRNPs, confirming our

GRADR predictions (Fig. 6A–C, Supplementary Fig. 8A and B, Supplementary Data 9). Subsequent hnRNP L UV-crosslinking and co-immunoprecipitation (UV-CLIP) experiments confirmed the interaction of ROCKI with hnRNP L (Fig. 6D, Supplementary Fig. 8C, D) - a protein previously implicated in immune-regulation[25]. Mass spectrometry analysis of hnRNP L interacting proteins further suggested that ROCKI may be part of a larger hnRNP complex, additionally comprising e.g. hnRNP A2B1 and hnRNP D, predicted by GRADR and ChIRP-MS to bind to ROCKI, respectively (Fig. 6D, E, Supplementary Fig. 8E, Supplementary Data 10). Binding motif analysis and established machine learning models (GraphProt[26] and RNAprot[27]) trained on publicly available hnRNP CLIP data furthermore revealed various potential binding sites for these hnRNPs in the ROCKI RNA sequence (Supplementary Fig. 9A, B). Beyond hnRNPs, ENCODE eCLIP data analysis substantiated the interaction of ROCKI with various proteins involved in splicing (Supplementary Fig. 9C), further supporting the suspected role for ROCKI in the nuclear mRNA processing machinery. While a report published during preparation of our manuscript suggested that ROCKI acts at the chromatin level, binding upstream of the *MARCKS* gene[24], our ChIRP-seq and -qPCR analysis did not detect ROCKI-binding at this site (Fig. 6F), supporting a role independent of direct chromatin interaction. Of note, it did, however, confirm the expected crosslinking of ROCKI to its own site of transcription, as a positive control for the ChIRP-seq assay (Fig. 6F). Closer examination of our ROCKI CRISPRi RNA-seq dataset revealed that the messenger of the GATA2 transcription factor was the most highly and significantly upregulated mRNA upon ROCKI silencing (Fig. 6G). Up-regulation of GATA2 mRNA upon ROCKI silencing could additionally be confirmed in siRNA-based experiments with G-MΦ and M-MΦ (Supplementary Fig. 9D). RT-qPCR analysis across all expressed exon-exon junctions of *GATA2* mRNA indicated that ROCKI specifically suppresses the maturation of the *GATA2* 3′ exons 5 and 6 (Fig. 6H, Supplementary Fig. 9E–G). GapmeR-mediated knockdown of several GRADR- and ChIRP-MS-predicted ROCKI-interacting hnRNPs identified hnRNPs A3 and A2B1 as being required for establishment of the *GATA2* mRNA exon 3-4 boundary (Fig. 6H, Supplementary Fig. 9E–G). Thus, ROCKI and hnRNP proteins appear to impact distinct *GATA2* mRNA exons, suggesting a model in which ROCKI piggybacks on an hnRNP complex required for specific *GATA2* mRNA processing steps, eventually enabling ROCKI to suppress the inclusion of 3′ exons and ultimately reduce *GATA2* mRNA abundance (Supplementary Fig. 9H). Investigating a direct connection between GATA2-inhibition and the pro-inflammatory activity of ROCKI, we force-expressed GATA2 via a lentiviral vector, thus overriding suppression by ROCKI. Intriguingly, proteomics analysis revealed that GATA2 transcription factor over-expression phenocopies ROCKI deficiency, entailing the suppression of the major pro-inflammatory mediators IL1b, CCL20, IL8, and PTGS2, alongside other ROCKI-regulated proteins (Fig. 6I, Supplementary Fig. 10A, Supplementary Data 11). When focusing on LPS-response genes, ROCKI silencing and GATA2 over-expression were found to increase the expression of several proteins associated with the interferon-response, while down-regulating pro-inflammatory factors (Fig. 6J, Supplementary Fig. 10B). Collectively, these findings reveal ROCKI as an hnRNP-associated, NFκB-inducible lncRNA that reinforces the inflammatory response by blunting a GATA2 transcription factor dependent anti-inflammatory program in macrophages (Fig. 6K).

Overall, our study establishes a mechanistic framework for understanding the roles of lncRNAs in human immunity, systematically charting their regulation, pathway dependencies and protein-interaction networks. By revealing RNA-driven control mechanisms in professional immune cells, our results position lncRNAs, such as ROCKI, LINC01215 and AC022816.1 as central players in coordinating innate immune responses. To accelerate progress in this emerging field, we made our comprehensive datasets available through the SMyLR web interface (rna-lab.org/smylr) (Fig. 6L and Supplementary

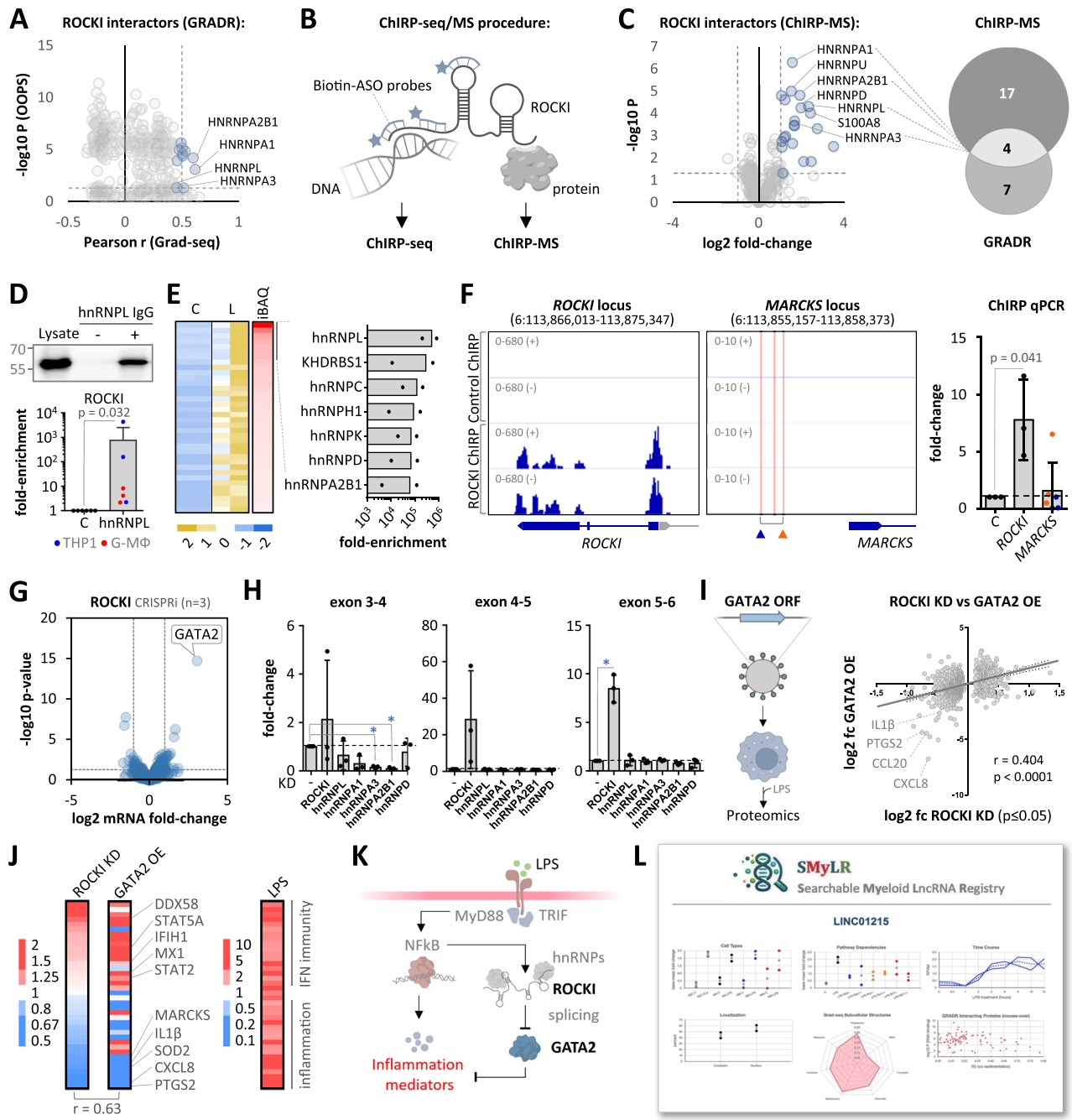

**Fig. 6 | Functional dissection of lncRNA ROCKI in macrophages. A** GRADR-derived interactome for ROCKI, analogous to Fig. 5E (two-tailed Student's test *p*-values and Pearson r). **B** Illustration of the ChIRP methodology. **C** Left: Volcano plot highlighting ChIRP-MS identified ROCKI interactors in THP1 cells (fold-changes comparing ROCKI to control ChIRP captures; two-tailed Student's test *p*-values). Right: Venn diagram showing overlap between GRADR and ChIRP-MS interactome predictions for ROCKI. **D** Top: Representative Western blot of hnRNP L in control (-) and hnRNP L ( + ) CLIP eluates (size marker positions in kDa indicated). Bottom: RT-qPCR analysis showing enrichment of ROCKI in hnRNP L CLIP eluates from THP1 and G-MΦ cells (two-tailed Student's t-test; three independent experiments; mean +/- SD). **E** Left: Row Z-score heatmap of protein abundance in two THP1 hnRNP L CLIP eluates (C = control, L = hnRNP L CLIP). Middle: iBAQ heatmap showing averaged total protein abundance in the hnRNP L CLIP. Right: Fold-enrichment plot for the top 7 eluted proteins. **F** Left: IGV plots showing control and ROCKI ChIRP-seq signals on both DNA strands at specified genomic loci (THP1 cells). Right: qPCR quantification of *ROCKI* and *MARCKS* locus enrichment in ROCKI- compared to control-ChIRP captures. MARCKS primer locations indicated by triangles below IGV plot (colors correspond to replicates shown in qPCR quantifications; three

independent experiments; mean +/- SD). Significant differences (*p* ≤ 0.05, One-way ANOVA test) indicated. **G** Volcano plot of mRNA regulation upon ROCKI silencing (data from Fig. 3C). Two-tailed Student's t-test *p*-values are shown. **H** RT-qPCR analysis of GATA2 mRNA levels in THP1 cells following the indicated ROCKI and hnRNP knockdowns (KD) (three independent experiments; mean +/- SD). Primers targeting the specified GATA2 exon-exon junctions were used. All significant changes (*p* ≤ 0.05, One-way ANOVA test) are denoted by blue asterisks. Exact *p*-values (in the same order as the asterisks): Exon 3-4: 0.004, 0.002. Exon 5-6: 0.032. **I** Left: GATA2 overexpression strategy. Right: Comparison of protein regulation upon ROCKI silencing (data from Fig. 4, two-tailed Student's t-test *p*-value ≤ 0.05)) versus GATA2 overexpression (THP1 cells). Pearson correlation analysis was performed, and r and *p*-value (two-tailed test) are shown. **J** Left: averaged fold-changes of LPS-inducible (≥2-fold induction) proteins upon ROCKI and GATA2 silencing (ranked by changes observed in the ROCKI silencing experiment). Right: Regulation of the respective proteins in response to LPS (data from Fig. 4D). **K** Summarizing model of ROCKI function in conjunction with GATA2 in macrophages. **L** Illustration of the SMyLR results page. All panels: G-MΦ treated with LPS for 4 h, THP1 cells for 8 h.

Fig. 11), providing an open-access resource for functional lncRNA exploration. We believe, our findings will facilitate future work aimed at deepening our understanding of RNA-mediated regulation in human health and disease.

## Discussion

Our findings suggest that lncRNAs exert more extensive control over cellular innate immune responses than previously thought. The five lncRNAs ROCKI, AC022816.1, LINC00158, LINC01215, and AC010980, investigated in this study, collectively influence the expression of at least 16% of the mRNAs regulated in the macrophage LPS response (Fig. 3E). These are complemented by additional immune-regulatory lncRNAs such as LUCAT1 and MaIL1[8–12]. Notably, the five lncRNAs in the focus of the present study were selected arbitrarily, implying that many of the remaining LPS-responsive lncRNAs may also have regulatory functions in macrophages, potentially influencing an even greater proportion of the immune-responsive transcriptome. Our LPS time-course RNA-seq analysis further supports that not all LPS-responsive lncRNAs operate at the same time but rather at various stages, with some, like MaIL1, acting early in TLR signaling, and others modulating the sustained immune response (Supplementary Fig. 3F). These observations underscore the complexity of lncRNA-mediated regulatory networks in macrophages and their potential to orchestrate distinct layers of innate immune control across different temporal and functional axes.

Our findings particularly underscore the role of ROCKI as a pro-inflammatory lncRNA in humans. Although we cannot rule out that ROCKI regulates chromatin, as previously speculated[24], we found no evidence supporting its postulated binding to the *MARCKS* gene region. Instead, ROCKI appears to inhibit *GATA2* mRNA expression (Fig. 6G, H). Expression of the GATA2 transcription factor is typically restricted to hematopoietic progenitor cells[28] and overexpression of GATA2 disrupts normal macrophage function, including the ability to express inflammation mediators (Fig. 6I–K) and to perform efferocytosis[29]. This suggests, that ROCKI helps maintaining the functional integrity of terminally differentiated macrophages by preventing the re-expression of progenitor state genes. Mechanistically, ROCKI appears to interact with hnRNP proteins and modulate *GATA2* mRNA processing. The hnRNPs A3 and A2B1, which appear to be part of a ROCKI-binding complex (Fig. 6A–E), promote the establishment of the *GATA2* exon 3-4 junction, while ROCKI itself appears to inhibit primarily the formation of the exon 5-6 junction (Fig. 6H). We hypothesize, that ROCKI is brought into proximity to *GATA2* mRNA via the hnRNP complex, where it may either shift the activity of this complex or recruit additional factors suppressing the maturation of *GATA2* mRNA 3' sequence elements. The precise mode of action of the ROCKI-hnRNP complex, potentially in cooperation with other RNA processing factors, remains to be elucidated. Furthermore, the specific GATA2 protein binding sites in the genome and epigenetic alterations that contribute to reduced pro-inflammatory activity of macrophages upon ROCKI silencing warrant further investigation.

Our study suggests that beyond ROCKI, numerous other lncRNAs participate in macrophage immunity. The introduction of GRADR marks a major advancement in identifying their interaction partners, a crucial step for lncRNA functional analysis. By integrating co-sedimentation, RNA-binding potential, and subcellular localization, GRADR enables the identification of potential interactors for a large number of lncRNAs in a single experiment. However, it does not confirm direct interactions, which is a key limitation compared to targeted affinity purification methods such as eCLIP or RAP-MS. The strengths and limitations of GRADR are reflected in our findings on the protein interactomes of LUCAT1 and ROCKI. GRADR identified only a fraction of the interactors detected by ChIRP-MS (~14-15% for LUCAT1 and ~19% for ROCKI) (Supplementary Fig. 6G, Fig. 6C). However, for LUCAT1, among these were interactors previously reported by others[10],

reinforcing the ability of GRADR to highlight relevant candidates. Given the interexperimental variability in RNA affinity purifications, as demonstrated for LUCAT1 (Supplementary Fig. 6G), we believe that this capacity of GRADR to refine likely interactors and pathways for further investigation is particularly valuable. As a stand-alone tool, GRADR is best suited for identifying broader cellular machineries (i.e. through pathway analysis) that lncRNAs associate with rather than selecting individual interactors. We recommend that interactor selection from GRADR predictions be complemented either by RNA affinity purification studies as discussed above, or alternative approaches, such as machine-learning models and eCLIP-based predictions, as demonstrated for ROCKI in this study (Supplementary Fig. 9B, C).

Interestingly, most of the immune-inducible lncRNAs catalogued by us seem to be poorly conserved beyond primates (Fig. 3A). This limited conservation might explain why regulators such as ROCKI or LINC01215 have been overlooked in traditional immunology studies focusing on murine models. The greater sequence flexibility of lncRNAs, which allows them to evolve new functions more rapidly compared to coding genes[30] might facilitate the development of new species-specific regulators, that tailor mammalian immune systems to diverse ecological niches. Thus, while rodent models are invaluable for elucidating fundamental principles of immunity, a thorough understanding of lncRNA functions may be crucial for fully grasping immune-associated mechanisms contributing to antimicrobial defense and diseases in humans.

As lncRNAs gain recognition as key players in immune-related diseases, including severe COVID-19, COPD, and IBD[9–12,24], future iterations of the SMyLR web interface could be expanded to incorporate genomic and expression data from diverse patient cohorts to assess clinical relevance. Although this study lays substantial groundwork, more research is required to fully elucidate the implications of lncRNAs in human immunity and disease. We anticipate that the resources and insights provided here will catalyze further work, advancing our understanding of the roles of lncRNAs in immune-regulation and opening new opportunities for disease biomarker discovery and therapeutic intervention.

## Methods
### Cell and tissue isolation and cultivation

All uses of human material were approved by the ethics committees of the University of Marburg Medical Faculty (UMR-FB20) or the Justus Liebig University of Giessen Medical Faculty (JLU-FB11). All recruited volunteers provided written informed consent. All work with primary human samples is described in detail below.

Alveolar type II epithelial cells (AECII) and precision cut lung slices (PCLS): For the isolation of AECII, healthy human lung tissue (e.g. healthy edge tissue from tumour resections; UMR-FB20 ethics approval number: Studie 224/12 and Studie 122/19) was sliced and washed with balanced salt solution buffer (BSSB) pH 7.4 (137 mM NaCl, 5 mM KCl, 0.7 mM $Na_2HPO_4$, 10 mM HEPES, 5.5 mM Glucose, 1.8 mM $CaCl_2$, 0.7 mM $MgSO_4$). Tissue was digested in 40 ml BSSB containing 2.86 mg trypsin (#T4799, Sigma) and 300 μl of 5 U/ml elastase (#LS002292, Worthington Biochemical) at 37 °C for 45 min while shaking. Digestion was stopped by transferring the tissue sample into 40 ml inhibition solution (30 ml DMEM/F-12 (#11320033, Gibco), 10 ml FCS (#S0615, Biochrom) and 1 ml DNaseI (10,000 U/ml, #4716728001, Sigma)) and cells were detached from the tissue by pipetting and passing the cell suspension successively through a 100 μm and 40 μm cell strainer (#431752, #431750, Corning), followed by centrifugation at 400 x g for 10 min. Cells were resuspended in adhesion medium (22.5 ml SAGM (#11645490, Lonza), 22.5 ml DMEM/F-12, 10% FCS, 0.5 ml DNaseI 10,000 U/ml), seeded and incubated at 37 °C for 2 h. The supernatant was collected and the adhesion step was repeated a total of three times, followed by centrifugation at 400 x g for 10 min. Cell pellet was resuspended in 2 ml SAGM medium. A discontinuous

Percoll gradient was prepared by pipetting 10 ml low density Percoll in a tube and layering 10 ml high density Percoll below. Resuspended cell pellet was layered on top followed by centrifugation at 400 x $g$, 20 min, 4 °C. Interface was collected, washed with PBS, and pelleted[31]. AECIIs were cultured for 7 days in SAGM supplemented with 2% FCS and 1% penicillin/streptomycin (#15140122, Thermo Fisher) on a 0.4 µm ThinCert transwell (#665641, Greiner Bio-One) at a density of 5 ×10⁵ cells / 113.1 mm², followed by stimulations and cell harvesting, as indicated. For the generation and stimulation of PCLS, lung tissue (JLU-FB11 ethics approval number: AZ31/93 and AZ58/15) with healthy appearance (e.g. tumor edge tissue) from patients undergoing surgery was processed. Tissue was filled with 3% low-melting agarose diluted in medium with PBS (1:1) and filled through the bronchi. Sections of 300 to 400 µm thickness were generated using a vibratome (Leica Biosystems). PCLS were cultured in 12-well dishes with DMEM, supplemented with GlutaMAX (#35050061, Thermo Fisher), 10% FCS, and 1:500 Normocin (#ant-nr-05, Invivogen) for 4 days and one additional day without antibiotics prior to further treatment.

Alveolar macrophages and patient BAL-pellets: Bronchoalveolar lavage (BAL) fluid (BALF) samples were collected from healthy donors or from patients who underwent BAL for pulmonary disease assessment at the University Clinics Giessen and Marburg. Each person, undergoing bronchoscopy including bronchoalveolar lavage (BAL), gave both oral and written informed consent. The study received approval from the local ethics committee (UMR-FB20 ethics approval number: Studie 87/12 and Studie 168/12). During bronchoscopy, a flexible fiberoptic bronchoscope was used, and BAL was performed by instilling 150 ml of pre-warmed sterile 0.9% NaCl solution. BAL was done in the right middle lobe or the lingula. For alveolar macrophage (aMΦ) isolation, BALF from healthy donors was centrifuged at 400 x $g$ for 10 min and cell pellet was washed once with PBS, followed by resuspension in X-Vivo 15 medium (#BEBP02-061Q, Lonza) supplemented with 5% FCS, 0.1% penicillin/streptomycin and 25 µg/ml gentamicin (#15750060, Gibco). Cells were then filtered through a 100 µm cell strainer. aMΦs were counted based on morphology, seeded at a density of 4 ×10⁵ cells/ml, and incubated for 2 h at 37 °C and 5% CO₂. Medium was then replaced with fresh medium and cells were stimulated as indicated and collected. BALF from patients (Supplementary Table 1) was centrifuged at 300 x $g$ for 10 min and cell pellet was snap-frozen in liquid nitrogen, followed by −80 °C storage and eventually RNA-isolation and RT-qPCR as described below.

Blood-derived and THP1 macrophages: Buffy coats were obtained from the Transfusion Medicine department, University Clinics Giessen and Marburg. Leukocytes were isolated using Lymphoprep (#18061, Stemcell Technologies) and MACS-purification (#130-050-201, Miltenyi CD14+ beads). Isolated monocytes were cultured for 7 days in X-Vivo 15 medium supplemented with 5% FCS and 15 ng / ml GM-CSF or 15 ng / ml M-CSF to obtain blood-derived macrophages (G-MΦ and M-MΦ, respectively). THP1 cells were maintained at a density of 2 × 105 – 1 ×106 cells / ml in RPMI-1640 (#31870074, Thermo Fisher), supplemented with 10% FCS, 1% sodium pyruvate (#11360070, Thermo Fisher), 1% GlutaMAX (Thermo Fisher) and 1% penicillin/streptomycin solution. THP1 cells were differentiated into macrophage-like cells with 80 nM phorbol 12-myristate 13-acetate (PMA) for 72 h.

Cell and tissue stimulations and cultivation conditions: If not specified otherwise in the figures, cells were treated with the following stimuli for the indicated durations: 1 µg/ml flagellin from *Salmonella enterica* serovar Typhimurium (FLA-ST, Invivogen), 100 ng/ml LPS from *Salmonella enterica* serovar Typhimurium, (BioXtra L6143, Sigma), 100 ng / ml recombinant human IFNα (rh IFNα 2 A Stemcell) or 200 ng/ml Pam3CSK4 (tlrl-pms, Invivogen). For PCLS stimulations 500 ng/ml LPS or FLA-ST were used. For bacterial infections, all bacteria were cultured in LB medium, with the exception of *S. pneumoniae*, which was cultured in THY medium. Cells were infected at a multiplicity of infection (MOI) of 10. For pathway inhibition, cells were

pre-treated with the respective inhibitor or DMSO for 2 h followed by further treatments. Inhibitors used were: BAY 11-7082 (NFκB and NLRP3 inflammasome inhibitor, #tlrl-b82, Invivogen; 1, 5 or 10 µM), Ruxolitinib (JAK-STAT inhibitor, #FBM-10-4511, Biozol; 0.1, 1 or 5 µM) and BX795 (TBK1/IKKε inhibitor, #tlrl-bx7, Invivogen; 0.1, 1 or 5 µM). All cells were maintained at 37 °C in a humidified atmosphere with 5% CO₂.

## Gene silencing and overexpression

Stable knockdown cell lines were generated using the CRISPR interference (CRISPRi) vector pLV-hU6-sgRNA-hUbC-dCas9-KRAB-T2a-GFP (Addgene #71237). Guide RNAs (gRNAs; Supplementary Table 2), targeting the transcriptional start site regions of genes of interest, were cloned into the CRISPRi vector, followed by lentiviral particle production and transduction as described below. Control cell lines were generated by transducing cells with the same empty CRISPRi vector, lacking a guide RNA insert. The transduced cells were sorted for GFP-expression (Aria III cell sorter, BD; Supplementary Fig. 12) and brought back into culture.

For LNA Gapmer and siRNA based gene silencing, 5 × 10⁵ THP1 cells or CD14+ monocytes (12-well format) were differentiated into macrophage-like cells as described above. After differentiation, medium was exchanged with antibiotic free medium. For transfection, 40 pmol antisense LNA GapmeR or 50 pmol SilencerSelect siRNA were complexed with 1.5 µl Lipofectamine 2000 (#11668019, Thermo Fisher) in 200 µl OptiMEM (#31985062, Thermo Fisher) and incubated for 15 min at room temperature. Liposomes were added to the cells, followed by centrifugation at 850 x $g$, 37 °C for 2 h. Macrophages were cultured for additional 40 h, followed by stimulations and cell harvesting. Knockdowns were verified by RT-qPCR. LNA GapmeRs and siRNAs used in the present study are listed below (Supplementary Table 3).

For generation of overexpression cell lines, the pLVX-M-puro lentivector (Addgene #125839) was used. The GATA2 coding sequence was amplified from a pLXV-zsGreen lentiviral plasmid containing the GATA2 coding sequence (a gift from Bryan Heit[29]). For PCR amplification the following primers were used:

5'-tccgatggatcccgATGGAGGTGGCGCCCGAGCAG-3 (forward, primer number OBS-4006)

5'-atcggagaattcCTAGCCCATGGCGGTCACCAT-3' (reverse, primer number OBS-3985)

The GATA2 coding sequence was cloned into the multiple cloning site of the pLVX-M-puro vector (BamHI and EcoRI), followed by lentiviral particle production and transduction of THP1 cells as described below. The empty pLVX-M-puro vector served as a control. Positive selection of transduced cells was achieved using puromycin (2 µg/ml, 3 days).

## Lentiviral transduction

Calcium phosphate transfection was used for co-transfection of 10 µg lentiviral vector, pseudotyping- and helper-plasmid (3.5 µg pVSVG and 6.5 µg psPAX2) into HEK293T cells. To this end, 3.5 × 10⁶ cells were seeded in a cell culture dish (100 × 20 mm) with 8.5 ml DMEM (#41965039, Gibco), 10% FCS and cultivated overnight, leading to a confluence of 70-85%. The plasmids, 106.3 µl 2 M CaCl₂ and 743.7 µl H₂O (minus plasmid volume) were mixed and 2x HBS was added dropwise while vortexing. The mixture was incubated for 10 min at room temperature, added to the cells and incubated at 37 °C, 5% CO₂ for 8 h. The cells were then washed with PBS and cultured in DMEM for 2 days. The supernatant containing the viral particles was passed through a 0.45 µM filter and concentrated overnight by ultra-centrifugation (Sorvall Discovery 90SE, Beckman Coulter Rotor 70 Ti) at 64,000 x g and 4 °C. 5 ×10⁵ cells were seeded in a 12-well and transduced by adding the viral pellet resuspended in RPMI-1640. Medium containing viral particles was exchanged after 2 days. 3-4 days

after transduction, cells were sorted for GFP expression (Supplementary Fig. 12) or selected with puromycin, as described above.

## Subcellular fractionation (RNA and Protein)

Cytoplasm/nucleus fractionations for RNA measurements were carried out by adding 200 μl lysis buffer to $5 \times 10^6$ cells per condition (10 mM Tris pH 8, 140 mM NaCl, 1.5 mM MgCl$_2$, 0.5% Igepal, 2 mM vanadyl ribonucleoside complex), followed by incubation on ice for 5 min. Lysate was centrifuged at 1000 x $g$, 4 °C for 3 min and the supernatant (cytosolic fraction) was transferred to a new tube. The pellet was the nuclear fraction. The cytosolic fraction was centrifuged again at 15,000 x $g$, 4 °C for 10 min and the supernatant was transferred to a new tube. The nuclear fraction was washed twice with 160 μl lysis buffer (centrifugations at 1000 x $g$, 4 °C for 3 min) and finally resuspended in 100 μl lysis buffer. RNA from both fractions was isolated with TRIzol (#15596026, Ambion) as described below. An equal amount of *Escherichia coli* K12 total RNA was spiked in before adjusting the concentrations of cytoplasmic and nuclear RNA for RNA-seq library preparation, described below.

Cytoplasm/nucleus fractionations for protein measurements were carried out by washing 5 ×106 cells per condition with PBS and collection of the cells by centrifugation at 1000 x $g$ for 8 min. Cells were then resuspended in 400 μl buffer A (10 mM Hepes pH 7.9, 10 mM KCl, 0.1 mM EDTA, 0.1 mM EGTA, 0.5 mM DDT, 1x cOmplete EDTA-free (#11836170001, Roche)), and incubated on ice for 15 min. Lysis was achieved by passing the cells through a 26 G needle attached to a 1 ml syringe (7-8 times). Upon centrifugation at 5000 x $g$ for 2 min, supernatant (cytosolic fraction) was transferred to a new tube. The pellet was the nuclear fraction. The cytosolic fraction was centrifuged at 20,000 x $g$ for 20 min and the supernatant was transferred to a new tube. The nuclear fraction was washed twice with buffer A and the nuclei pellet was resuspended in 100 μl buffer B (20 mM Hepes pH 7.9, 400 mM NaCl, 1 mM EDTA, 1 mM EGTA, 0.5 mM DDT, 1x cOmplete EDTA-free (Roche)), followed by 1 h incubation on a shaker at 4 °C. To obtain a pure nuclear extract, the obtained lysate was centrifuged at 20,000 x $g$ for 20 min and the supernatant was transferred to a new tube. Proteins were precipitated from the cytosolic and nuclear fractions by adding three times the volume of ice cold acetone, followed by overnight incubation at −20 °C. Proteins were pelleted by centrifugation at 17,000 x $g$ and 4 °C for 30 min, air-dried and resuspended in 50 μl 8 M Urea, 0.1 M ammonium-bicarbonate.

## ChIRP-seq and -qPCR

DNA antisense probes for ChIRP (Supplementary Table 4) were synthesized at Metabion AG. For 3' mono-biotinylation, terminal transferase (#M0315, New England Biolabs) and Biotin-11-ddUTP (#NU-1619-BIOX-L, Jena Bioscience) were used according to the manufacturer's instructions. $2 \times 10^7$ cells were used per capture and ChIRP-seq was performed as described previously[32]. Briefly, to preserve RNA–chromatin interactions, cells were cross-linked with 20 ml of 1% glutaraldehyde in PBS for 10 min at room temperature while shaking. Cross-linking was stopped by adding 1/10$^{th}$ volume of 1.25 M glycine and incubating for 5 min. Cells were pelleted ($2000 \times g$, 5 min), washed with ice-cold PBS, flash-frozen in liquid nitrogen, and stored at −80 °C. Frozen pellets were thawed, weighed, and lysed in 300 μl buffer (50 mM Tris-Cl pH7.0, 10 mM EDTA, 1% SDS, 1× cOmplete EDTA-free (Roche), 1 mM PMSF (#93482, Sigma-Aldrich), 200U/ml murine RNase inhibitor (#M0314, NEB)). Lysates were sonicated using a Bioruptor Plus (Diagenode) at high intensity (30 s on, 45 s off, 90 min) in 15 ml tubes. Fragmentation (100–500 bp) was verified by treating 5 μl of lysate with 90 μl proteinase K (PK) buffer (100 mM NaCl, 10 mM Tris-Cl pH8.0, 1 mM EDTA, 0.5% SDS, 5% [v/v] Proteinase K (#AM2546, Ambion)), followed by DNA purification (GeneJET PCR Kit, #K0701, Thermo Scientific), and analysis on a 1% agarose gel. Sonicated lysates were transferred to 2 ml tubes, adjusted to 1 ml final volume with lysis buffer, flash-frozen in liquid nitrogen, and stored at −80 °C. For hybridization, samples were thawed; 10 μl aliquots were put aside as RNA and DNA inputs. The remaining sample was split in half and each portion mixed with 2 ml hybridization buffer (750 mM NaCl, 1% SDS, 50 mM Tris-Cl pH7.0, 1 mM EDTA, 15% formamide, 1× cOmplete EDTA-free, 1 mM PMSF, 200 U/mL RNase inhibitor). Biotinylated DNA probes (100pmol/ml chromatin) were added and incubated at 37 °C for 4 h with shaking. C-1 streptavidin beads (100 μl per 100pmol probe, #65001, Invitrogen) were washed thrice with 1 ml lysis buffer, resuspended in 100 μl lysis buffer, added to the lysate, and incubated for 30 min at 37 °C with shaking. Beads were washed five times with 1 ml pre-warmed wash buffer (2× SSC, 0.5% SDS, 10 mM PMSF), including 5-min incubations at 37 °C. After the final wash, beads were resuspended and separated into 100 μl for RNA and 900 μl for DNA isolation.

For RNA isolation, 95 μl RNA PK buffer (100 mM NaCl, 10 mM Tris-Cl pH7.0, 1 mM EDTA, 0.5% SDS, 5% [v/v] Proteinase K) was added to the input aliquot and 100 μl to the bead-bound samples. After incubation at 50 °C for 45 min and denaturation at 95 °C for 10 min, samples were chilled on ice. RNA was isolated and qPCR was performed as described below.

For DNA extraction, beads were resuspended in 150 μl and input aliquot in 140 μl DNA elution buffer (50 mM NaHCO$_3$, 1% SDS, 100 μg/mL RNase A). Samples were incubated at 37 °C for 30 min with shaking, and supernatants were collected using a magnetic stand. The extraction was repeated and pooled. Proteinase K (15 μl, 20 mg/ml) was added, followed by incubation at 50 °C for 45 min. DNA was purified via phenol:chloroform:isoamyl extraction and used for high throughput sequencing and RT-qPCR.

## ChIRP-MS

DNA antisense probes used for ROCKI and LUCAT1 ChIRP-MS are shown in Supplementary Table 4. Probes were 3'-monobiotinylated as described above. $2 \times 10^8$ cells were used per capture. ChIRP-MS was performed as described previously[33], with the following modifications: (1) no RNase-treatment control; (2) proteins were on-bead digested and eluted from the beads as specified in the mass spectrometry section. Briefly, to preserve RNA–protein interactions, $4 \times 10^8$ cells were cross-linked in 3% formaldehyde in PBS for 30 min at room temperature while shaking. Cross-linking was stopped by the addition of 0.125 M glycine and incubation for 5 min. Cells were pelleted by centrifugation at $2000 \times g$ for 5 min, washed once with ice-cold PBS, and centrifuged again. The supernatant was removed, and the pellet was weighed, flash-frozen in liquid nitrogen, and stored at −80 °C until further processing. Frozen pellets were dissolved in cell lysis buffer (50 mM Tris-Cl pH7.0, 10 mM EDTA, 1% SDS, 1× cOmplete EDTA-free protease inhibitor cocktail (Roche), 1 mM PMSF (Sigma-Aldrich), and 200 U/mL RNase inhibitor (NEB)) and sonicated using a Bioruptor Plus (Diagenode) at high intensity (30 s on / 45 s off) until the lysate became clear. DNA fragmentation was assessed by mixing 5 μl of lysate with 90 μl PK buffer (100 mM NaCl, 10 mM Tris-Cl pH8.0, 1 mM EDTA, 0.5% SDS), followed by centrifugation at $16,000 \times g$ for 10 min. The resulting pellet and supernatant were treated separately with 5 μl Proteinase K (5% v/v; Ambion) and incubated at 50 °C for 45 min. DNA was purified using the GeneJET PCR Purification Kit (Thermo Scientific), and the fragment size (1–2 kb) was verified by 1% agarose gel electrophoresis. Lysates were flash-frozen and stored at −80 °C.

For hybridization, thawed lysates were aliquoted (10 μl) for RNA and protein input controls and incubated at 37 °C. The remaining lysate was divided equally. To pre-clear the lysate, 30 μl of C-1 streptavidin magnetic beads were washed twice with 100 μl lysis buffer, incubated with the lysate at 37 °C for 30 min with shaking, and removed using a magnetic stand. The supernatant was transferred to a new tube and this step was repeated to remove any residual beads.

Hybridization was carried out by adding 2 volumes of hybridization buffer (750 mM NaCl, 1% SDS, 50 mM Tris-Cl pH7.0, 1 mM EDTA, 15% formamide, 1× cOmplete EDTA-free (Roche), 1 mM PMSF, and 200 U/mL RNase inhibitor) and 1 µl of 100 µM biotinylated DNA probes per 1 ml of lysate. Samples were incubated at 37 °C for 16 h with shaking. Prior to use, 100 µl of C-1 streptavidin beads per 1 µl probe were washed three times with 1 ml lysis buffer. Following hybridization, 1 ml of the lysate was mixed with the washed beads and incubated for 30 min at 37 °C with shaking. Beads were then separated on a magnetic stand and washed five times with 1 ml of wash buffer (2× SSC, 0.5% SDS, 10 mM PMSF) at 37 °C, with 5 min incubation per wash. The final wash was performed using 2× SSC to remove residual detergents.

Beads were then separated magnetically, with 100 µl retained for RNA isolation and the remaining fraction used for protein analysis. RNA was extracted as described in the ChIRP-seq section, and proteins were processed as detailed in the mass spectrometry protocol.

## (RT-)qPCR

Total RNA was isolated using TRIzol (Ambion) and contaminating gDNA was removed by treatment with DNaseI (Thermo Fisher) in the presence of recombinant murine RNase inhibitor (NEB), followed by PCI extraction. RNA concentration was determined using a Nanodrop 2000 spectrometer (Thermo Scientific). cDNA was generated (#4368813, High-Capacity cDNA Reverse Transcription Kit, Thermo Fisher), and RT-qPCR was performed using Luna Universal qPCR Master Mix (#M3003, NEB). For RNA-quantifications in subcellular fractionation and ChIRP-capture experiments, the Luna Universal One-Step RT-qPCR Kit (#E3005, NEB) was used. For DNA quantifications in ChIRP-capture experiments, the Luna Universal qPCR Master Mix (NEB) was used. All (RT-)qPCRs were performed on a QuantStudio 3 instrument (Applied Biosystems). Relative expression changes were calculated using the $2^{-\Delta\Delta CT}$ method[34], based on CT values and using U6 snRNA as a housekeeper control, unless stated otherwise. For BAL patient samples, equal amounts of RNA were reverse transcribed, followed by RT-qPCR analysis as described above. However, CT values were not processed using the $2^{-\Delta\Delta CT}$ method. Instead, log2 $2^{-\Delta CT}$ values were calculated for the lncRNAs shown in Fig. 1I, as well as for U6 snRNA and RPS18 mRNA as controls (Supplementary Fig. 2D), using patient # 3 (the sample with the highest IFNB1 CTs), as a reference. These values were then plotted against IFNB1 log2 $2^{-\Delta CT}$ values, followed by linear regression and Pearson correlation analysis. Primer oligos used for RT-qPCR are listed in Supplementary Table 5.

## High Throughput Sequencing

RNA was isolated using TRIzol reagent as described above and handed over to the in-house genomics core facility (Philipps-University Marburg, medical faculty) or Genewiz GmbH for library generation and sequencing. RNA quality was validated (Agilent RNA 6000 Nano Kit). Illumina Truseq stranded total RNA libraries with rRNA depletion (subcellular fractionation experiments) or Lexogen QuantSeq 3' mRNA-Seq FWD libraries (all remaining RNA-seq experiments) were generated. Libraries were paired end sequenced with 2 × 150 bp (Truseq libraries) or single end sequenced with 75-bp read length (Lexogen libraries) on a NextSeq 550 instrument. ChIRP-seq stranded genomic DNA libraries were generated at Vertis Biotech AG (Freising, Germany) using in-house protocols and sequenced on a NexSeq 550. External RNA-seq data (Supplementary Table 6) were downloaded from the NCBI GEO repository using the SRA tool 'prefetch'. Downloaded.sra files were converted to.fastq files using the SRA tool 'fastq-dump'.

Demultiplexed Fastq files were imported into the CLC Genomics Workbench v 10.0.1. Sequence reads were trimmed using the CLC workbench with quality score limit set to 0.05 and 25 nt minimal read length. Trimmed sequences were mapped to the GRCh38 human reference genome using the CLC workbench with the following

parameters: Mismatch cost: 2. Insertion cost: 3. Deletion cost: 3. Length fraction: 0.8. Similarity fraction: 0.8. Max. hits for a read: 10. Strand specific: both. The subcellular fractionation RNA-seq data were additionally normalized to *Escherichia coli* spike-in RNA. To this end, the human GRCh38 (GENCODE) and *E. coli* K12 MG1655 (NCBI:GCF_000005845.2_ASM584v2) reference annotation- and genome-files were fused and upon read mapping (with the same parameters as above), separate tables of uniquely mapping reads were created for both organisms. Human RPKMs were calculated based on the human read table and subsequently normalized to the percentage of *E. coli* reads in each sample. For all other RNA-seq experiments, normalized gene expression values (e.g. RPKM) were calculated based on the raw read counts obtained (uniquely mapping reads). To focus on reliably detectable RNAs, all RNA-seq data tables were filtered to include only RNAs with RPKM values ≥ 0.5 in all experimental replicates under at least one of the compared conditions. This filtering criterion, combined with a minimal number of 10 sequencing reads detected under at least one condition, was applied uniformly in Fig. 1, where the lncRNAs central to this study were identified. In subsequent analyses, the lncRNAs included in Fig. 1E were exempted from this filter to ensure their consistent inclusion in all downstream analyses, while all other RNAs continued to be subject to the original filtering criteria. For experiments where RNA-seq data were obtained in triplicates, DeSeq2[35] was executed in R Studio to calculate expression changes and *p*-values. For visualization of read coverages, mapped read data were exported in BAM format and files were analysed using the Integrated Genomics Viewer (IGV[36]).

## Orthogonal organic phase separation (OOPS)

$3 \times 10^6$ monocytes isolated from buffy coats were seeded into a 90-mm dish, differentiated, and stimulated with LPS as described above. OOPS was conducted as described[37]. Protein from the input, eluate (RNase-treatment), and control-elution (no RNase treatment) samples was precipitated using acetone, collected by centrifugation, and air-dried. Pellets were resuspended in 8 M Urea, 0.1 M ammonium-bicarbonate for mass spectrometry analysis, as described below. A detailed protocol for the OOPS procedure is provided in Supplementary Note 1.

## Protein mass spectrometry

For ChIRP-MS and CoIP-MS analysis, capture beads were transferred to the mass spectrometry facility, and proteins were digested on-bead and eluted as follows: Samples were washed three times with 100 µl ammonium-bicarbonate (50 mM, pH 8.0) each. Trypsin (0.1 µg in 50 µl ammonium-bicarbonate buffer) was added to the beads and samples were incubated at 37 °C for 45 min. The supernatant was transferred into fresh tubes and digested over-night at 37 °C. For reduction of disulphide bridges 5 mM DTT (Dithiothreitol) was added. Samples were then incubated for 15 minutes at 95 °C. Subsequently, the sulfhydryl-groups were chemically modified by adding iodoacetamide to a final concentration of 25 mM and incubating samples for 45 min at room temperature (RT) in the dark. Excess iodoacetamide was quenched by the addition of 50 mM DTT and incubation for one more hour at RT.

For whole proteome analysis, $1 \times 10^6$ cells were differentiated and stimulated as described above. Cells were washed once with PBS, pelleted by centrifugation, snap frozen, and stored at -80 °C till further use. For proteomics analysis, the cells were lysed (8 M Urea, 0.1 M Ammoniumbicarbonate) and sonicated (Bioruptor Plus, Diagenode) with 2 × 10 seconds on, 30 seconds off and intensity setting "high". The lysate was cleared by centrifugation and protein concentration was determined using the BCA method (#23225, Pierce™ BCA Protein Assay Kit, ThermoFisher and an Infinite PRO (Tecan) plate reader). Samples were adjusted to equal protein concentrations and mass spectrometry analysis was done. Specifically, to each sample solubilized in 8 M Urea buffer, 1.25 µl of a 0.2 M TCEP solution in 0.1 M

NH$_4$HCO$_3$ was added and samples were incubated for 1 h on a shaker at 37 °C and 1000 rpm. After cooling to 25 °C, 1.25 µl of 0.4 M iodoacetamide in bidest water was added. Subsequently, samples were incubated in the dark at 25 °C and 500 rpm. Then, 1.25 µl of a 0.5 M N-acetyl-cystein solution in 0.1 M NH$_4$HCO$_3$ was added and samples were incubated for 10 min at 25 °C and 500 rpm. If necessary, pH was adjusted to 8-9. Then, the urea concentration was decreased to 6 M urea by dilution with 0.1 M NH$_4$HCO$_3$ and 2.5 µl of a 0.2 µg / µl LysC solution in 0.1 M NH$_4$HCO$_3$ were added. Samples were then incubated at 37 °C for 3 h. Subsequently, samples were further diluted using 0.1 M NH$_4$HCO$_3$, resulting in a final urea concentration of 1.6 M. Then, 4 µl of 0.5 µg / µl Sequencing Grade Modified Trypsin (Serva) in 0.1 M NH$_4$HCO$_3$ were added. Samples were incubated at 37 °C over-night. TFA was added to a final concentration of ~1% to yield a pH below 2. Samples were centrifuged at 14,000 rpm.

Both for "On-Bead digestion" samples and samples solubilized in 8 M urea buffer, the resulting peptides were desalted and concentrated using Chromabond C18WP spin columns (Macherey-Nagel, Part No. 730522), according to the manufacturer's protocols. Finally, peptides were dissolved in 25 µl of water with 5% acetonitrile and 0.1% formic acid. Peptide concentration was measured using a Nanodrop (Thermo Scientific) and samples were diluted to 100 ng of peptides per µl. Mass spectrometric analysis of the samples was performed using a timsTOF Pro mass spectrometer (Bruker Daltonic). A nanoElute HPLC system (Bruker Daltonics), equipped with an Aurora 25 cm × 75 µm C18 RP column filled with 1.7 µm beads (IonOpticks) was connected online to the mass spectrometer. A portion of approximately 200 ng of peptides corresponding to 2 µl was injected directly on the separation column. Sample loading was performed at a constant pressure of 800 bar. Separation of tryptic peptides was achieved at 60 °C column temperature with either a 30 min gradient or a 100 min gradient of water / 0.1% formic acid (solvent A) and acetonitrile/0.1% formic acid (solvent B) at a flow rate of 400 nl / min. 30 min gradient: Linear increase from 2% B to 17 % B within 18 minutes, followed by a linear gradient to 25% B within 9 min and linear increase to 37% solvent B in additional 3 min. Finally, B was increased to 95% within 10 min and hold at 95% for additional 10 min. 100 min gradient: Linear increase from 2% B to 17% B within 60 min, followed by a linear gradient to 25% B within 30 min and linear increase to 35% solvent B in additional 10 min. Finally, B was increased to 95% within 10 min and hold for additional 10 min. The built-in "DDA PASEF-standard_1.1sec_cycletime" method developed by Bruker Daltonics was used for mass spectrometric measurement. Data analysis was performed using MaxQuant (different versions) with Andromeda search engine against the human Uniprot database. Peptides with a minimum of seven amino-acid lengths were used and FDR was set to 1% at the peptide and protein level. Protein identification required at least one razor peptide per protein group and label free quantification (LFQ) algorithm was applied. Perseus[38] was used for further evaluation of MaxQuant results, including contaminant removal and calculation of protein fold-changes and significances (two-tailed Student's t-test), based on LFQ and iBAQ abundance values. For ChIRP-MS data analysis, the average LFQ values of control samples were artificially set to 100 when all values in this group were zero, to enable inclusion of proteins exclusively detected in the lncRNA capture samples for volcano plot visualizations (Fig. 6C and Supplementary Fig. 6E).

## GRADR
For GRADR analysis of RNA-protein interactions, Grad-seq data[8], OOPS-MS data and subcellular fractionation RNA-seq and MS data were combined into one tabular dataset (Supplementary Data 8). For RNAs annotated as cytoplasmic (Supplementary Data 7), the GRADR dataset was filtered for RNA-binding proteins (OOPS-MS RNase + vs − elution fold-change ≥ 2, p-value ≤ 0.05 [Student's t-test]) detectable in the cytoplasm (proteins detected in the cytoplasmic fractions in all three independent replicates and not being ≥ 60% nuclear in all three replicates). Pearson correlation analysis was conducted, comparing the Grad-seq co-sedimentation profile of the RNA in question to the Grad-seq profile of each protein present in the dataset after applying the filtering steps described above. The dataset was sorted in descending order by the Pearson r values and entries with r values ≥ 0.5 were considered as potential protein-interactors of the RNA in question. For nuclear RNAs (Supplementary Data 7), the procedure described above was carried out analogously for nuclear proteins and additionally, mitochondrial proteins were removed from the dataset due to the implausible interaction of nuclear and mitochondrial factors. A detailed step-by-step protocol for the GRADR procedure is provided in Supplementary Note 1.

## UV crosslinking & co-immunoprecipitation
Co-IP was performed with 2 × 10$^7$ cells per capture according to a previously published procedure[39], with minor modifications. UV-crosslinking was carried out with 300 mJ / cm$^2$ (HL-2000 HybriLinker) and the protein of interest was isolated from the cell lysate with capture antibody-coated protein G dynabeads (#10003D, Thermo Fisher). Per Co-IP, 2.5 µg of hnRNP L- and control-IgG antibodies were used. RNA was isolated by boiling the beads, followed by PCI extraction and 30:1 ethanol / 3 M sodium acetate precipitation. For Co-IP with subsequent mass spectrometry analysis, the capture dynabeads were washed with 50 mM HEPES, pH 7.5 and resuspended in 0.1 M ammonium bicarbonate and further processed (mass spectrometry core facility, Philipps-University Marburg) as described above. Antibodies used for Co-IP experiments are listed below (Supplementary Table 7).

## Western blotting
Co-IP samples were separated on a 10% polyacrylamide gel by SDS-PAGE. For Western blotting, proteins were transferred onto nitrocellulose (Amersham Protran; Sigma-Aldrich; GE10600003) by semi-dry blotting. Blots were blocked with TBST buffer containing 5% milk powder and 3% BSA. Detection antibodies were used at a 1:10,000 (anti-hnRNP L) or 1:1000 (anti-mouse-HRP) dilution. Blot development was carried out using the ECL Prime Western Blot Detection kit (GERPN2232, Amersham) and a Chemostar Imager (INTAS Science Imaging). Western blot full scans can be found in Supplementary Fig. 8.

## FACS-based ELISA
Cytokine content in cell culture supernatants was detected using the LEGENDplex human inflammation panel 1 multiplex FACS kit (BioLegend, Cat: 740809, Lot No. B347171). The experiment was performed according to the manufacturer's protocol, using a BD FACSymphony A1 flow cytometer and BD FACS diva 9.0 software. Results were analyzed using the LEGENDplex data analysis software suite.

## LncRNA conservation analysis
Conservation analysis was conducted on macrophage lncRNAs identified in Fig. 1E as being up-regulated ≥ 2-fold in response to LPS. ENSEMBL-annotated MANE transcript variants were used, or alternatively the transcript variants best matching macrophage RNA-seq coverage. To avoid biases associated with coding sequence conservation, lncRNA regions overlapping protein-coding genes were clipped. Sequence conservation was assessed using NCBI BLASTN, comparing human lncRNA sequences against the main RefSeq reference genomes of the species included in Fig. 3A. BLAST hits with ≥ 20 complementary nucleotides were retained, and the sequence window with the longest query sequence coverage (within a maximum 100 kb region) was considered the orthologous locus. If BLAST hits mapped to multiple loci, syntenic regions were prioritized. Percent base conservation was calculated from all BLAST hits within this window. For genus-level generation lengths (Fig. 3A), the available data for all species within each genus were averaged[40]. Evolutionary distances

between species were inferred using data from the TimeTree database (http://www.timetree.org/home).

## Computational prediction of ROCKI-protein interactions

To identify potential hnRNP binding sites on the ROCKI transcript ENST00000434296, we used a combination of searching for known binding sequence motifs and predicting binding sites by machine learning models trained on publicly available CLIP data.

For the sequence motif search, two types of motifs were used: sequence motifs in MEME motif format[41] (visualized as sequence logos) or regular expressions/strings (for hnRNP C and hnRNP D). Sequence motifs were extracted from catRAPID omics v2.0 and ATtRACT curated motif databases[42,43]. MEME motif search was conducted using FIMO[44] as part of the RBPBench package (https://github.com/michauhl/RBPBench), with a set $p$-value threshold for reporting motif hits of 0.0002. The threshold was chosen such that for each RBP ≥ 1 hit was reported. Regular expression search was conducted with RBPBench.

To computationally predict hnRNP binding sites, we used publicly available eCLIP data from ENCODE[45] to train predictive models for two methods: GraphProt and RNAProt[26,27]. Data was available for hnRPN C and hnRNP L, while for the remaining hnRNPs there was either no eCLIP data available (hnRNP A2B1, hnRNP A3, hnRNP D) or not enough recovered binding sites for efficient training (hnRNP A1). IDR peak region files were downloaded from ENCODE and used as positive instances for training, while the negative instances were generated using RNAProt. Both GraphProt and RNAProt were trained using default settings (using only sequence features), with variable sequence length input and an up- and downstream site extension of 10 nt. For RNAProt, hits after applying an internal threshold (--thr 1) were used, while for GraphProt we used the top 10 predicted sites ranked by GraphProt score.

To search for further ROCKI binding proteins, we examined the overlap of the ROCKI full-length transcript with ENCODE available eCLIP data[45] using bedtools intersect −s −wao. Identified peaks were ranked by $p$-value, as reported in Supplementary Fig. 9C.

## Statistical analysis and data visualization

Statistical analysis of RNA-seq and proteomics data with ≥ 3 experimental replicates was performed using Deseq2 and Perseus (two-tailed Student's t-test), respectively, as described above. Regulations ≥ 2-fold with $p$-values ≤ 0.05 were considered statistically significant. Further data (e.g. RT-qPCR measurements) were analyzed in GraphPad PRISM and two-tailed Student's t-test was used for two-sample comparisons, and One-way ANOVA test was used for multiple-sample comparisons. $P$-values are written out in the figure panels, where possible, or where not possible, due to space limitations, statistical significance is indicated with an asterisk. Where appropriate, individual measurement points are shown. Else, error bars are provided, illustrating the standard deviation (SD). For heatmap visualization of RNA-seq and proteomics data, hierarchical clustering was done using Cluster 3.0[46] and heatmaps were generated using JAVA TreeView[47]. Pathway enrichment analysis was performed using ConsensusPathDB[48] or ENRICHR[49], as indicated. GO-term UMAPs (Fig. 4C) were generated using ENRICHR and processed as follows: For genes up- and down-regulated upon knockdown of each lncRNA, the associated "Biological Process" GO-terms were highlighted within the pre-computed ENRICHR GO-term UMAP obtained by following the provided Appyter link. For each lncRNA, the top 10 GO terms associated with the up- and down-regulated gene sets were identified. These GO-terms were marked on the UMAP, and lines were drawn in a vector graphic overlay to connect each GO-term to the corresponding lncRNA. In this representation, each lncRNA is depicted as a white circle with a black outline, positioned next to its associated GO-terms in a way allowing clear visualization of all connections. For UMAP representation of sedimentation similarities of cellular protein machineries and lncRNAs

(Fig. 5H), the "PCA, t-SNE, UMAP" Appyter (https://appyters.maayanlab.cloud/)[50] was used. The input data table included sedimentation patterns across the 22 Grad-seq fractions, expressed as row-Z-scores[8]. These data were provided for the indicated lncRNAs and diverse proteins classified as components of the specified protein machineries (Supplementary Data 12). Venn diagrams were generated using *Venn Diagram Generator* (http://barc.wi.mit.edu/tools/venn/) or *Venny 2.1* (https://bioinfogp.cnb.csic.es/tools/venny/). PCA analysis was performed in R Studio, based on RPKMs using the R-script prcomp (stats) and the rgl package. Network plots were generated using Cytoscape version 3.7.2. Violin plots were generated using BoxPlotR[51]. Other plots were generated in GraphPad PRISM or Excel.

## Reporting summary

Further information on research design is available in the Nature Portfolio Reporting Summary linked to this article.

## Data availability

The high throughput sequencing data generated in this study have been deposited in the NCBI GEO database and are available under the accession codes GSE268546 (ChIRP-seq data; https://www.ncbi.nlm.nih.gov/geo/query/acc.cgi?&acc=GSE268546) and GSE268547 (RNA-seq data; https://www.ncbi.nlm.nih.gov/geo/query/acc.cgi?acc=GSE268547). The mass spectrometry proteomics data generated in this study have been deposited in the ProteomeXchange Consortium PRIDE[52] repository and are available under the accession code PXD061457: Key omics data obtained as part of this study are available as supplementary data files and through the SMyLR web interface (rna-lab.org/smylr). All data are included in the Supplementary Information or available from the authors, as are unique reagents used in this Article. The raw numbers for charts and graphs are available in the Source Data file whenever possible. Source data are provided with this paper.

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

## Acknowledgements

This project was supported by the DFG SFB-TR-84 (subproject C1 and C10) and the Hessian Ministry of Science and Art (HMWK) funding programs LOEWE Diffusible Signals (LOEWE/2/13/519/03/06.001(0002)/74) (subproject D2 and B1) (to L.N.S and B.S., respectively) and LOEWE Exploration (LOEWE/5/A004/519/06/00.005(0008)/E31) (to L.N.S). Parts of the work presented here were funded by the Fritz Thyssen Foundation (project 10.21.2.024MN) (to L.N.S.) and through the Federal Ministry of Education and Research (BMBF) projects PerMed-COPD (01EK2203A) and Deep Legion (031L0288A) (both to B.S.). M.U. was funded by Deutsche Forschungsgemeinschaft (DFG) grant BA 2168/25-1, TRR 167/2 NeuroMac, and GRK 2344/2 MeInBio. This study was supported by the German Research Foundation (DFG) under Germany's Excellence Strategy (CIBSS - EXC-2189 - Project ID 390939984) (to R.B.) and DFG SFB-1213 (Project A01, to S.S.P.). We would like to thank Kerstin Hoffmann for assisting in cell culture and Western Blot experiments and all patients and clinical services at UMR medical center.

## Author contributions

N.S., H.J., M.Aznaourova and L.N.S. designed the experiments; N.S., H.J., M.Aillaud., J.H., S.W., M. Aznaourova, H.S., L.D.H., F.B. and L.N.S. performed experiments; M.U., T.S., A.N., L.J., A.K., C.R., O.R., E.N., A.M., C.V., U.L., R.B., S.S.P. and B.S. contributed new reagents/tissue samples/ analytic tools; N.S., H.J., M.Aillaud, J.H., M. Aznaourova, L.D.H., M.U. and L.N.S. analyzed data; N.S. and L.N.S. wrote the paper; funding acquisition: L.N.S.; project supervision: L.N.S.

## Funding

## Competing interests

The authors declare no competing interests.
