## [Transparent Peer Review file · Nature Communications]

A searchable atlas of pathogen-sensitive lncRNA networks in human macrophages

Corresponding Author: Professor Leon Schulte

Version 0:

Reviewer comments:

Reviewer #1

(Remarks to the Author)

This study developed the GRADR method, which integrates glycerol gradient profiling with RNA binding proteome analysis to map the protein interactomes of all expressed RNAs, and revealed the important role of long non-coding RNAs in regulating human immunity. The research identified several lncRNAs (such as LINC01215, AC022816, and ROCK1) that play regulatory roles in macrophage immunity, and constructed the lncRNA-protein interaction network, providing valuable resources for lncRNA research. However, the authors should take writing seriously to avoid a large number of inconsistencies between the figures and descriptions of results. Additionally, I have some other comments that the authors should revise.

1. It is recommended that the authors provide a detailed description of the data preprocessing process in the manuscript. Specifically, before performing the differential expression analysis of RNA-seq data, authors should clarify the preprocessing steps involved in filtering and normalizing gene expression. This processing can help eliminate low-quality RNA that may arise from sample sourcing and technical limitations.
2. The authors should explain why a conservation analysis of lncRNAs was first performed in result 3, as the conservation of lncRNAs is not further elaborated upon in the current results.
3. Given that all of the authors' experiments were conducted in macrophages, it is recommended to change the title of Result 4 from "in immune cells" to "in macrophages."
4. The authors should explain why LINC01215 was considered an immune "negative" regulator in macrophages and why ROCK1 was considered as a potential "positive" regulator. In other words, they should clarify how "positive" and "negative" regulators are defined.
5. In Figure 5H, the authors used UMAP to perform dimensionality reduction on the correlation between proteins and lncRNAs, constructing a UMAP figure for the proteins. How did the authors further project the lncRNAs onto this UMAP figure?
6. The conclusions of some results are inconsistent with the figures. (1) Result 3 states that "AC010980.1 significantly influenced the global LPS response at the mRNA level," while Fig. 4D shows that the P value of AC010980.1 was 0.254. (2) Result 5 claims that "ROCK1 specifically suppresses the inclusion of the GATA2 3' exons 5 and 6," while Fig. 6H shows that the effects of ROCK1 on GATA2 3' exons 3-4, 4-5, 5-6 were all significant. (3) Result 5 mentions that "hnRNPs A3 and A2B1 are required for the establishment of the GATA2 mRNA exon 3-4 boundary," whereas Fig. S9E shows that there was no significant change in GATA2 after hnRNPs A2B1 knockdown. It is suggested that the authors carefully check the consistency between the figures and the text.
7. Some figures in the manuscript are not described in the main text, such as Fig. 3B, Fig. 4C, and Fig. 6J. Additionally, Fig. 4B was incorrectly referred to as Fig. 4F in the text. In addition, it is recommended to unify the name "ROCK1" in the text and in Fig. 4E.
8. Is it possible to obtain the corresponding binding proteins for specific lncRNA (e.g., lncRNA-protein, one-to-one) using the

GRADR method? If so, the authors should provide a detailed description of the specific steps to obtain the lncRNA-protein pairs.

9. It is recommended that the authors provide a more detailed description of GRADR in the Methods section, such as how the RNA-binding proteins were filtered out.

10. The authors should discuss the advantages and disadvantages of GRADR compared to other methods for identifying RNA-binding proteins.

11. Figures 2B and 2C, among others, lack significance annotations. The authors should carefully review the entire manuscript to ensure that the corresponding significance levels are properly indicated.

Reviewer #2

(Remarks to the Author)

Nils Schmerer et al. described pathogen-sensitive lncRNA networks in human macrophages. The authors integrated Grad-seq, OOPS-MS, CRISPR interference, and cyto/nuclei composition analysis and comprehensively provided candidate genes and proteins that are regulated by or interact with these immune-induced lncRNAs.

This study's workload is tremendous. It proves the anti-bacterial and pro-inflammatory role of lncRNA, ROCK1, in macrophages. The figures presented in this study are clear and of good quality, but I have major conceptual reservations regarding the findings and the study design.

Major concerns:

1. Though ex vivo induction of G-M Φ and M-M Φ is a commonly used approach to study the nature of macrophage, it is elusive why the authors focus on the G-M Φ instead of M-M Φ . Figure S1E-F does not support that 'Comparative analysis furthermore highlighted that G-M Φ , in particular, show a response pattern more akin to that of aM Φ , justifying their selection as our primary in vitro model for lncRNA investigations in this study (Fig. S1E-F)'. It seems that the M-M Φ are of the same similarity in the response pattern to that of aM Φ . In addition, only ~14% (11.4%+2.9%) overlapped up-regulated genes suggested a notable discrepancy between G-M Φ and aM Φ , instead of 'justifying their selection'.
2. The following study is based on the selected 5 lncRNAs presented in Figure 1F and 1H. However, this result is not convincing for the reason in the first major concern.
3. The authors repeatedly used the word 'unprecedented' to overstate their findings in this study, which is not unprecedented. The role of lncRNAs in regulating the immune response was reviewed ten years ago (PMID: 25113636). The authors should avoid using this kind of words to describe their work.
4. The whole study starts by using the primary M Φ isolated from the healthy donors to simulate the response under the pathogen attack, while gene regulation or lncRNA expression are not supported using external pathogen-related datasets. At least, the authors shall use external or in house-generated single cell RNA-seq data or any other form of data to support their key discoveries.
5. Since none of the five lncRNAs is located in the protein-coding gene, CRISPR knockout experiments instead of CRISPR interference might make the result more pronounced.
6. The authors introduced a new strategy by combining Grad-seq and OOPS-MS to dissect better RNAs and proteins which physically interact. To benchmark this strategy, looking into some reported lncRNAs, e.g., MAIL1, PACER, and LUCAT1, will be better. Whether or not these lncRNA and their interacting proteins can be found through GRADR.
7. Figure 6A-C, by ChIRP-MS cross-validation, only a small fraction of ROCK1 interactors were recovered by GRADR (4 out of 21); the other 7 interactors were found by GRADR less further validation. I think it is better to perform a third experiment (e.g., ChIRP-WB), which would help readers evaluate the power of GRADR.

Minor points:

1. The SMyLR cannot be accessed without password.
2. Two small figures on the right side of Fig. 2B are poorly described.
3. Names of lncRNA used in the manuscript should be consistent, e.g., MROCK1/ROCK1.
4. Fig.3C and 3D, the authors reported that a series of mRNA/lncRNA are up/down regulated in the CRISPRi experiment. Did the authors use lncRNA over expression to validate this finding?

Reviewer #3

(Remarks to the Author)

This study describes the development of GRADR which is a gradient based RNA interactor profiling tool. The goal is to develop a novel method allowing for the capture of global lncRNA protein interaction maps within macrophages. This is accompanied by a web user interface which makes this tool highly accessible especially to researchers that perhaps lack bioinformatic skills. Controls used are sound and mechanistic insights into ROCK1 as a confirmation of the tool in addition to the other lncRNA candidates is notable and provides confidence in the tool.

There is a large amount of omics data available through this manuscript including RNA-seq data from various cell types, stimulations and proteomics. This data will be a welcome resource to the macrophage community and I only have a small number of suggestions and clarifications to make conclusions clearer.

- Details on what GRADR actually does and the measures of success are very vaguely discussed in the manuscript. They

mention that positive controls work which is great but providing additional numbers on how many false positives are found and what type of threshold should be set in order consider something a possible hit for follow up studies would be beneficial. Also after I took a quick look through the online web interface I looked up some of their example lncRNAs and all have a top hit of KHDRBS1, can the authors comment on why this protein which is presumably a false positive comes up at the top hit.

- CHIRP on ROCK1, appears to be only a single replicate comparing direct CHIPR to control from GSE268546 . Can the authors comments on this
- GSE269547, the token provided to the reviewer did not work and therefore I was unable to evaluate this data
- Table S1: For all CRISPRi experiments it is unclear what the control is in each case. Are they using multiple nontargeting controls for the experiments? This is also not clear in the figure legends for Figure 3 and 4

Minor point

- Fig 6C. A sized venn diagram is preferable

Version 1:

Reviewer comments:

Reviewer #1

(Remarks to the Author)

The authors have taken care to address the comments. There is no remaining comments.

Reviewer #2

(Remarks to the Author)

Reviewer #3

(Remarks to the Author)

The authors have made the suggested revisions and it is a much improved manuscript

Point-by-point response

We sincerely appreciate the instructive feedback on our manuscript provided by all three reviewers. In response, we have made substantial revisions, including **additional experiments** (e.g., ChIRP studies, lncRNA expression quantifications in M-M Φ and in patient samples, ROCK1 knockdowns in primary M Φ), **data analyses** (e.g., analysis of external RNA-seq datasets), **text and figure refinements**, addition of a **GRADR step-by-step protocol** and other updates. These revisions, prompted by the reviewers' suggestions, have helped us refine our study and further improve the manuscript, as detailed below.

All text changes made during the revision process are indicated in blue in the revised manuscript, with the exception of the newly added GRADR step-by-step protocol (which uses blue colour to highlight the sections). Additionally, text and figure modifications are outlined in the following point-by-point responses, detailing how each reviewer comment has been addressed and incorporated into the manuscript.

Reviewer #1 (Remarks to the Author)

This study developed the GRADR method, which integrates glycerol gradient profiling with RNA binding proteome analysis to map the protein interactomes of all expressed RNAs, and revealed the important role of long non-coding RNAs in regulating human immunity. The research identified several lncRNAs (such as LINC01215, AC022816, and ROCK1) that play regulatory roles in macrophage immunity, and constructed the lncRNA-protein interaction network, providing valuable resources for lncRNA research. However, the authors should take writing seriously to avoid a large number of inconsistencies between the figures and descriptions of results. Additionally, I have some other comments that the authors should revise.

Response: We thank the reviewer for recognizing the significance of our study in identifying key lncRNAs involved in macrophage immunity and for acknowledging the value of the GRADR-derived lncRNA-protein interaction network as a resource for the community. In response to the reviewer's concerns, we have carefully reviewed the entire manuscript and revised figure legends, the Results text, and statistical annotations to ensure clarity and accuracy. Additionally, we have addressed all specific comments and made corresponding improvements throughout the manuscript.

1. It is recommended that the authors provide a detailed description of the data preprocessing process in the manuscript. Specifically, before performing the differential expression analysis of RNA-seq data, authors should clarify the preprocessing steps involved in filtering and normalizing gene expression. This processing can help eliminate low-quality RNA that may arise from sample sourcing and technical limitations.

Response: In response to the reviewer's comment, we have now updated the description of the RNA-seq data preprocessing steps in the Methods section, documenting the RPKM cutoffs applied in our analysis. We apologize for the initial omission of this important information. Following read quality trimming and mapping (as described in the manuscript), we routinely apply RPKM cutoffs to eliminate genes that are hardly detected, thereby reducing background noise and enhancing the stringency of our analysis. Consistent with our prior publications, we applied an RPKM cutoff of 0.5, which had to be surpassed in all replicates under at least one experimental condition. We have found this threshold to constitute an effective balance between filtering out noise and retaining lowly expressed, but potentially functional, transcripts.

The **updated Methods section** under “High Throughput Sequencing” (**page 14, last paragraph**) now details how this RPKM cutoff was applied across all datasets, with only one exception: the lncRNAs central to this study, identified in Fig. 1E, were exempted from the RPKM cutoff starting from Fig. 2 to ensure their consistent appearance in all downstream analyses. This exemption was necessary to prevent sporadic exclusion of some of these lncRNAs in the figure Panels due to RPKM values dropping below 0.5 in certain replicates. Outside of this exception, the cutoff was strictly maintained. Additional filtering criteria, such as fold-change thresholds and p-value cutoffs, are described in detail in the Results section and figure legends, as well as in the newly added GRADR step-by-step protocol, now included in the Supplementary Information.

2. The authors should explain why a conservation analysis of lncRNAs was first performed in result 3, as the conservation of lncRNAs is not further elaborated upon in the current results.

Response: In the current study, the conservation analysis serves to provide additional context for the five lncRNAs central to this work. We believe this analysis helps narrow down the possibilities for studying these molecules in animal models and offers insights into the evolutionary role of these lncRNAs in mammalian immunity. We have **revised the relevant Results section** to clarify the approach and significance of these results, **and moved it to the section “lncRNAs are embedded in different pathways and phases of the immune response”**, along with the results on macrophage infections with bacterial pathogens (Fig. 2E), for improved logical flow (**page 5, lines 30-40**). The conservation analysis are also discussed further in the pre-last discussion paragraph. Additionally, we have **updated the methods section (new paragraph “lncRNA conservation analysis”, page 17)**, providing relevant details on the conservation analysis procedure which were still missing.

Generally, we believe that the conservation of immune-associated lncRNAs across mammals, including those reported here and previously published, demands a more comprehensive investigation and discussion than possible in the current study. We are currently preparing a separate manuscript that systematically examines lncRNA conservation. This upcoming work incorporates Nanopore-based expression data from macrophages across diverse species and explores the evolutionary conservation of LPS-responsive lncRNAs in detail. We believe that this in-depth analysis is beyond the scope of the present study and best presented as an independent publication.

3. Given that all of the authors' experiments were conducted in macrophages, it is recommended to change the title of Result 4 from "in immune cells" to "in macrophages."

Response: We thank the reviewer for this comment and agree - the statement “in immune cells” has been changed to “in macrophages”.

4. The authors should explain why LINC01215 was considered an immune "negative" regulator in macrophages and why ROCK1 was considered as a potential "positive" regulator. In other words, they should clarify how "positive" and "negative" regulators are defined.

Response: We agree that the definitions of "positive" and "negative" regulators in the manuscript need clarification. Specifically, ROCK1 was considered a potential "positive" regulator because its knockdown led to reduced expression of genes up-regulated during the macrophage LPS-response, suggesting its role in enhancing immune activation. Conversely, LINC01215 and AC022816.1 were identified as "immune negative" regulators because their knockdown resulted in an opposite impact on the LPS-response, pointing to a role in dampening excessive inflammation. We revised the manuscript to explicitly define these terms and clearly link them to the functional evidence provided by our experiments. We believe that in the **revised Results paragraph (page 6, lines 22-29)** describing Fig. 4D, the definition of "positive" and "negative" regulators is now more understandable for the readers.

5. In Figure 5H, the authors used UMAP to perform dimensionality reduction on the correlation between proteins and lncRNAs, constructing a UMAP figure for the proteins. How did the authors further project the lncRNAs onto this UMAP figure?

Response: We thank the reviewer for pointing this out. We acknowledge that the description of the UMAP-based analysis was incomplete in our manuscript, both for Fig. 5H and Fig. 4C. To address this, we have **added the missing information in the Methods section under the revised heading “Statistical analysis and data visualization.” (page 18)**. Furthermore, we have **revised the Fig. 4 and 5 legends and the Fig. 5H Results text (page 7, lines 29-30)** to improve clarity. As detailed in the revised Methods section, in Fig. 5H, the lncRNAs were not separately projected onto the UMAP but were included as part of the input dataset (they were additionally highlighted by adding black outlines, as now mentioned in the figure legend). This dataset comprised the sedimentation patterns of all components along the Grad-seq gradient, represented as row Z-scores. To further enhance transparency, we have provided this **input dataset for the Fig. 5H UMAP as a new supplementary dataset (“Dataset S12”)**.

6. The conclusions of some results are inconsistent with the figures. (1) Result 3 states that “AC010980.1 significantly influenced the global LPS response at the mRNA level,” while Fig. 4D shows that the P value of AC010980.1 was 0.254. (2) Result 5 claims that “ROCK1 specifically suppresses the inclusion of the GATA2 3' exons 5 and 6,” while Fig. 6H shows that the effects of ROCK1 on GATA2 3' exons 3-4, 4-5, 5-6 were all significant. (3) Result 5 mentions that “hnRNPs A3 and A2B1 are required for the establishment of the GATA2 mRNA exon 3-4 boundary,” whereas Fig. S9E shows that there was no significant change in GATA2 after hnRNPs A2B1 knockdown. It is suggested that the authors carefully check the consistency between the figures and the text.

Response (1): We thank the reviewer for pointing out this inconsistency. **In the Results text describing Fig. 4D, we inadvertently referred to AC010980.1 when we actually meant AC022816.1**, which indeed shows a significant impact at the RNA level. This error **has been corrected** in the revised text.

Response (2-3): We appreciate the opportunity to clarify these points, as there seems to be a failure on our side to communicate these results. While GATA2 mRNA levels appear elevated upon ROCK1 knockdown when using primers targeting the exon-exon junctions 3-4, 4-5, and 5-6, statistical significance was only observed for the 5-6 junction. Regarding the hnRNPs, only for the hnRNPA3 and A2B1 knockdowns we observed a significant effect on GATA2 mRNA levels. This effect was observed when using primers targeting the exon-exon junction 3-4, but not with primers probing the other exon-exon junctions, as shown in Fig. 6H and Fig. S9F-G. Thus, these hnRNPs and ROCK1 regulate specific, yet distinct GATA2 mRNA processing events. This led us to the following conclusion, **now rephrased in the results for additional clarity**: *“Thus, ROCK1 and hnRNP proteins appear to impact distinct GATA2 mRNA exons, suggesting a model in which ROCK1 piggybacks on an hnRNP complex required for specific GATA2 mRNA processing steps, eventually enabling ROCK1 to suppress the inclusion of 3' exons and ultimately reduce GATA2 mRNA abundance” (page 8, lines 23-26)*. Which additional factors are potentially involved in the suppression of these exons by ROCK1 remains to be determined, as we now emphasize in the **revised discussion (page 9, second paragraph)**.

We have made the following additional revisions:

- We **changed the colour of all asterisks denoting significance from black to blue** to distinguish them from data points and reduce potential misinterpretation.
- We added a **new panel, Fig. S9F, displaying the GATA2 mRNA exon structure** and the positions of the primers used. Additionally, we **extended panel E to show the ROCK1 knockdown validation** for this experiment and specified in **panel G** that the exon 1-2 junction of GATA2 mRNA was not detectable in RT-qPCR.

- The legends for Fig. 6 and Fig. S9 have been updated accordingly.

We hope these changes enhance the clarity of the presented results and make them easier to interpret.

7. Some figures in the manuscript are not described in the main text, such as Fig. 3B, Fig. 4C, and Fig. 6J. Additionally, Fig. 4B was incorrectly referred to as Fig. 4F in the text. In addition, it is recommended to unify the name "ROCKi" in the text and in Fig. 4E.

Response: We thank the reviewer for highlighting these points and have **made the necessary corrections**:

- Figures 3B, 3E, 4C, and 6J are now explicitly described in the Results text (page 6, first paragraph; page 6, line 20; page 8, line 31-33), and the figure legends for Fig. 2B, 3B, 4C, and 6J have been revised to add missing information.
- The incorrect reference to "Fig. 4F" in the text has been corrected to "Fig. 4B and C" (page 6, line 20).
- The name ROCKI has been unified throughout the manuscript, including in Fig. 3C, Fig. 4E, and the Methods section. Additionally, in Fig. S9B, we have replaced "MROCKI-201" with "ROCKI", while also providing its transcript stable ID (ENST00000434296) for clarity. We also reviewed the entire manuscript to make sure all other gene names have unified spellings.

8. Is it possible to obtain the corresponding binding proteins for specific lncRNA (e.g., lncRNA-protein, one-to-one) using the GRADR method? If so, the authors should provide a detailed description of the specific steps to obtain the lncRNA-protein pairs.

Response: We appreciate the reviewer's question. While GRADR can predict individual lncRNA-protein interactions, we believe this should not be its primary application. Its primary strengths lie in:

A) Narrowing down the global lncRNA-protein interaction landscape to systematically categorize lncRNAs based on their associated protein-machineries.

B) Refining lncRNA-protein interaction predictions from targeted approaches such as ChIRP-MS, RAP-MS or SHIFTR, as demonstrated for ROCKI in our study. All these approaches are inherently noisy and GRADR may filter out implausible interactions, thus adding more resolution.

To enhance transparency, we have **added a step-by-step protocol** at the end of the Supplementary Information file, outlining how GRADR can be used to prioritize potential interactors. Additionally, the **SMYLR database now offers the possibility to download a list of GRADR-predicted interactors for each lncRNA** (link provided below the "GRADR-Predicted Interactors" plot on the results page). However, we emphasize that this list should primarily be used for narrowing down larger protein machineries rather than individual proteins a given lncRNA may interact with and for filtering candidates from targeted affinity purification experiments. We have **further clarified this point in the Discussion (page 9, last paragraph)**.

9. It is recommended that the authors provide a more detailed description of GRADR in the Methods section, such as how the RNA-binding proteins were filtered out.

Response: We thank the reviewer for this suggestion. As mentioned in our reply above, we have now added a **detailed step-by-step protocol for GRADR** (at the end of the Supplementary Information file), including all critical data processing steps.

10. The authors should discuss the advantages and disadvantages of GRADR compared to other methods for identifying RNA-binding proteins.

Response: Our original discussion briefly addressed this aspect, but in response to the reviewers' feedback we have now expanded and refined this section.

In particular, we have **revised the discussion to incorporate new data (e.g., Fig. S6D-G), which further illustrate the strengths and limitations of GRADR (page 9, last paragraph)**. While GRADR provides a global, transcriptome-wide overview of plausible RNA-protein interactions by integrating co-sedimentation, RNA-binding potential, and subcellular localization, it does not directly confirm interactions, making it less specific than targeted affinity purification approaches such as eCLIP or RAP-MS. However, we emphasize that GRADR is particularly valuable for refining interactors from targeted affinity purification experiments, helping to filter out background noise and prioritize proteins for further validation.

We believe these refinements provide a more in-depth discussion of the potential applications, trade-offs, and optimal use cases for GRADR and we thank the reviewer for prompting this improvement.

11. Figures 2B and 2C, among others, lack significance annotations. The authors should carefully review the entire manuscript to ensure that the corresponding significance levels are properly indicated.

Response: We have carefully reviewed the manuscript to ensure that statistical significance annotations are properly indicated. In response, we have **introduced the following updates**:

- **Fig. 1H**: Statistical significance annotations added, and figure legend and text updated.
- **Fig. S1B**: Significance annotations added, and the legend revised accordingly.
- **Fig. 2B-D**: Statistical significance annotations added and legend updated; in panel C, the vertical axis was changed to a log scale to improve result interpretation and enhance visibility of significance markers.
- **Fig. S3B, C, D, G**: Statistical significance annotations added, and the legend revised accordingly.
- **Fig. S4A**: Statistical significance annotations added, with corresponding legend updates.
- **Fig. 6D, F and H**: Significance annotations added, and the legend revised accordingly.
- **Fig. S8B**: Statistical significance annotations added, and the legend updated.
- **New Fig. S9D and E**: statistical significance annotations added; legend and Results text updated accordingly. In panel E, the ROCK1 knockdown validation, which was still missing, was added.

Reviewer #2 (Remarks to the Author)

Nils Schmerer et al. described pathogen-sensitive lncRNA networks in human macrophages. The authors integrated Grad-seq, OOPS-MS, CRISPR interference, and cyto/nuclei composition analysis and comprehensively provided candidate genes and proteins that are regulated by or interact with these immune-induced lncRNAs.

This study's workload is tremendous. It proves the anti-bacterial and pro-inflammatory role of lncRNA, ROCK1, in macrophages. The figures presented in this study are clear and of good quality, but I have major conceptual reservations regarding the findings and the study design.

Response: We appreciate the reviewer's recognition of the comprehensive nature of our study and the extensive experimental work involved. We have carefully addressed the reviewer's concerns by revising the manuscript to clarify our study design and further ensure that our conclusions are well-supported by the data. Among others, we have added additional validation experiments (e.g., lncRNA expression analyses in M-M Φ and patient cohort samples, external RNA-seq data analysis, ROCK1 knockdowns in primary G-M Φ and M-M Φ), refined our interpretation of key findings, and strengthened both the Results and Discussion sections to better articulate the rationale behind our approach and the significance of our findings. We believe these revisions, prompted by the reviewer's insightful feedback and detailed below, have significantly enhanced the clarity and depth of the manuscript.

Major concerns:

1. Though ex vivo induction of G-M Φ and M-M Φ is a commonly used approach to study the nature of macrophage, it is elusive why the authors focus on the G-M Φ instead of M-M Φ . Figure S1E-F does not support that 'Comparative analysis furthermore highlighted that G-M Φ , in particular, show a response pattern more akin to that of aM Φ , justifying their selection as our primary in vitro model for lncRNA investigations in this study (Fig. S1E-F)'. It seems that the M-M Φ are of the same similarity in the response pattern to that of aM Φ . In addition, only ~14% (11.4%+2.9%) overlapped up-regulated genes suggested a notable discrepancy between G-M Φ and aM Φ , instead of 'justifying their selection'.

Response: We thank the reviewer for raising this important point and acknowledge that the previous version of the manuscript did not clearly convey our initial aims and conclusions from the macrophage profiling experiments. The issue likely arose when we shortened the manuscript, inadvertently omitting relevant details. Our primary goal was not to mimic alveolar macrophages (aM Φ) but rather to identify both differences and common core lncRNA programs shared across different macrophage types that are likely critical for antimicrobial defence. Our data highlight both distinct and overlapping responses of G-M Φ , M-M Φ , and aM Φ to bacterial stimuli, but, as the reviewer correctly points out, they do not indicate a clear preference for either G-M Φ or M-M Φ as a model. Ultimately, we selected G-M Φ as our primary in vitro model due to their scalability and availability (human aM Φ from healthy donors are hardly available) and the essentiality of GM-CSF for human lung macrophage replenishment and pulmonary defence (now cited in the revised manuscript). However, we fully acknowledge that both G-M Φ or M-M Φ recapitulate core macrophage lncRNA responses (including those seen in aM Φ) and **for the revised manuscript, we have expanded our data analysis and performed further experiments** to support this view. To address this point, we have done the following revisions:

- **Results text (page 4, pre-last paragraph): revised to clarify our rationale** for selecting G-M Φ while recognizing the suitability of M-M Φ and to incorporate new data detailed below.
- **Fig. S2A (left): Updated heatmap showing lncRNA regulation (≥ 2 -fold in both replicates) across AECII, aM Φ , G-M Φ , and now also M-M Φ .**

- **Fig. S2A (right):** New bar diagram illustrating how **lncRNAs upregulated in aMΦ** are **similarly regulated in G-MΦ and M-MΦ**.
- **New RT-qPCRs and analysis of external RNA-seq datasets**, as well as **ROCK1 silencing experiments in G- and M-MΦ**, confirming a shared core lncRNA response and shared lncRNA functions across macrophage types, as further **detailed in response to major point 2 and 4 and minor point 4**.

We believe these revisions better communicate our rationale for focusing on G-MΦ while acknowledging the similar suitability of M-MΦ for investigating the lncRNAs in focus.

2. The following study is based on the selected 5 lncRNAs presented in Figure 1F and 1H. However, this result is not convincing for the reason in the first major concern.

Response: We thank the reviewer for this comment. As detailed above, the five lncRNAs selected for this study are part of a core program of the human macrophage response, shared across different macrophage types. In addition to the new RNA-seq data visualizations presented in Fig. S2A the **new Fig. S3G (time-series RT-qPCR analysis) confirms that the selected lncRNAs are regulated not only in aMΦ and G-MΦ, but also in M-MΦ**, with the exception of LINC01215, whose regulation appears restricted to aMΦ and G-MΦ. Furthermore, we now show that the core function of ROCK1, in regulating GATA2 expression, is recapitulated in knockdown experiments using both G-MΦ and M-MΦ, as explained in response to minor point 4.

To further substantiate the relevance of the five lncRNAs in focus to human innate immunity, we **re-analyzed external RNA-seq datasets (detailed in our reply to point 4), which confirmed the robust regulation of the five lncRNAs** in human macrophages and monocytes in response to LPS (**new Fig. S2B-C**). Moreover, as shown in **new Fig. 1I and Fig. S2D-E**, we **quantified the expression of these lncRNAs in bronchoalveolar lavage (BAL)-derived cells from 24 human patients** undergoing BAL for pulmonary disease assessment. Our new results show that all five lncRNAs exhibit a significant linear correlation with IFNB1, which we previously characterized as a reliable marker of pulmonary immune activation in BAL studies (PMID: 32241891). Notably, this correlation - supporting their involvement in pulmonary immunity in vivo - was not observed for other RNAs, used as controls (U6 snRNA and RPS18 mRNA; Fig. S2D-E).

To incorporate these new data, supporting the relevance of the lncRNAs in focus beyond the G-MΦ model, we have updated the manuscript as follows:

- **Fig. 1I and Fig. S2D-E: new RT-qPCR experiments**, demonstrating the correlation of the five lncRNAs with IFNB1 in **BAL samples from 24 human patients**. The BAL procedure, RT-qPCR strategy and patient cohort are documented in the updated Methods (page 11, second paragraph; page 14, first paragraph; new Table 1). The results are described on **page 5, top paragraph**.
- **Fig. S2B-C: Re-analyzed external RNA-seq data** demonstrating the ≥ 2 -fold up-regulation of the five lncRNAs with varying significances in macrophage and monocyte stimulation experiments conducted by other labs (updated methods text on page 14, lines 28-30 and new Table 6; updated Results text on **page 4, line 41-43**).
- **Fig. S3G: New RT-qPCR experiments**, determining the **regulation of the five selected lncRNAs in M-MΦ**, as well. The results are described on **page 5, line 28-30**.
- The figure legends for Fig. 1, Fig. S2 and Fig. S3 have been revised accordingly.

We believe these updates illustrate a broad engagement of the selected lncRNAs in human macrophage immune responses and pulmonary defense, further justifying their selection in this study.

3. The authors repeatedly used the word ‘unprecedented’ to overstate their findings in this study, which is not unprecedented. The role of lncRNAs in regulating the immune response was reviewed ten years ago (PMID: 25113636). The authors should avoid using this kind of words to describe their work.

Response: We agree with the reviewer that the term “unprecedented” is not appropriate in this context. While we believe that our study provides substantial new insights into the roles of lncRNAs in human macrophages and provides as a rich resource for the community, we acknowledge that the role of lncRNAs in immune regulation has been recognized and explored previously. We have **revised the manuscript to ensure that the language reflects the novelty of our findings without overstating them**. Terms such as ‘novel’ and ‘unprecedented’ have been removed throughout the manuscript.

4. The whole study starts by using the primary MΦ isolated from the healthy donors to simulate the response under the pathogen attack, while gene regulation or lncRNA expression are not supported using external pathogen-related datasets. At least, the authors shall use external or in house-generated single cell RNA-seq data or any other form of data to support their key discoveries.

Response: We appreciate the reviewer’s suggestion to benchmark the lncRNA regulations highlighted in our study against external macrophage RNA-seq datasets, as this is an important step in assessing the robustness and recognizing potential biases in our experiments. To address this, we searched the NCBI GEO database for suitable datasets and selected **three different macrophage and monocyte RNA-seq profiling studies**, each containing three experimental replicates (**datasets documented with accession numbers and Pubmed IDs in Table 6**).

We prioritized datasets where cells had been stimulated with LPS for durations comparable to our study (4 hours). While we did not find enough datasets with this exact time point, we identified three published studies where cells were stimulated with LPS for 6 hours (in one case, with additional IFN γ co-stimulation), which we considered suitable proxies. Across all datasets, we observed **consistent upregulation of the five lncRNAs in the focus** of our study, with an average fold-change ≥ 2 across the three replicates of each dataset (**new Fig. S2B**). In one dataset (Managò et al.), we observed greater variability between replicates, causing fluctuations in fold-change values, preventing some lncRNAs from meeting the $p \leq 0.05$ threshold. However, in the other two studies (Lewis et al. and Lissner et al.), all five lncRNAs met both the fold-change and statistical significance criteria (**new Fig. S2C**). We believe these external data provide additional support for the relevance of the lncRNA regulations observed in our study to human monocyte and macrophage immunity.

We have revised the **Results text (page 4, line 41-43)** to document the regulation of the lncRNAs in these external datasets and **updated the Methods section** (updated text on page 14, lines 28-30 and new Table 6) and Fig. S2 legend accordingly.

5. Since none of the five lncRNAs is located in the protein-coding gene, CRISPR knockout experiments instead of CRISPR interference might make the result more pronounced.

Response: We thank the reviewer for this valuable suggestion. We have previously used Cas9-based lncRNA knockout strategies and published a small methods paper on ncRNA knockouts with minimal genomic perturbation (PMID: 29451908) to address concerns such as large deletions disrupting intra- and inter-chromosomal interactions. However, we have since **shifted away from knockout strategies due to potential additional artifacts introduced by the process of clonal expansion**. Cas9-based knockouts require single-cell clonal expansion for proper characterization, and we have observed that this process alone can create artifacts: even unmodified single-cell clones from the same cell line display varying LPS responses, potentially due to chromosomal instability and heterogeneity within immortalized cell lines. Further issues with this procedure, due to clonal cell expansion have been reported by others (e.g. PMID: 36307508). To avoid these issues, we now use CRISPR interference (CRISPRi), which does not require clonal expansion and reduces this source of bias.

While we agree that lncRNA gene knockouts could potentially result in stronger phenotypic effects, in the light of the above-mentioned concerns, **we sought to rather substantiate our key findings using independent methods, not relying on CRISPR-Cas proteins.** As **detailed in our response to minor point 4**, we now silenced ROCK1 through RNAi in primary G-M Φ and M-M Φ and could reproduce the upregulation of *GATA2* mRNA, identified as the most pronounced expression change in the ROCK1 CRISPRi experiments, supporting the robustness of our results (new Fig. S9D).

6. The authors introduced a new strategy by combining Grad-seq and OOPS-MS to dissect better RNAs and proteins which physically interact. To benchmark this strategy, looking into some reported lncRNAs, e.g., MAIL1, PACER, and LUCAT1, will be better. Whether or not these lncRNA and their interacting proteins can be found through GRADR.

Response: We appreciate the reviewer's suggestion and have benchmarked GRADR against published interactome data for immune-associated macrophage lncRNAs. We focused on LUCAT1 due to its well-characterized role as an innate immune regulator with known interactors. We **compared GRADR predictions of the LUCAT1 interactome (new Fig. S6D)** with targeted **LUCAT1 ChIRP-MS affinity purifications**, either **previously published (Vierbuchen et al., PNAS 2023; PMID: 36577072; Review Fig. 1A) or newly conducted by us (new Fig. S6E-G)**. Vierbuchen et al. identified LUCAT1 as a splicing regulator, associating with mRNA-processing factors, such as hnRNPs, SFPQ and DDX17. Importantly, our new LUCAT1 ChIRP-MS data confirm the association of LUCAT1 with hnRNPs, DDX17, and nuclear speckle components and generally with splicing related pathways (new Fig. S6E-F).

RNA affinity purification methods are inherently noisy, which is also illustrated by the limited overlap between our LUCAT1 ChIRP-MS data and those of the Vierbuchen et. al. study (although both highlight the association with key splicing-related factors) (Fig. S6G). This variability complicates the selection of key interactors for functional studies, which is where GRADR provides an advantage. Notably, our **GRADR analysis supports the ChIRP-MS predicted role of LUCAT1 in splicing** (Fig. S6D and F). Overlaying ChIRP-MS data with our GRADR predictions highlighted **LUCAT1 interactions with hnRNPs and nuclear speckle components** (NONO predicted by GRADR, NONO and SFPQ by ChIRP-MS; Fig. S6D-E). The association of LUCAT1 with nuclear speckle proteins is supported by SFPQ CLIP-seq data, which we recorded for another unpublished study and cannot be integrated into the current manuscript (**Review Fig. 1B**). This result underscores the **ability of GRADR to highlight both confirmed and novel plausible interactions in noisy affinity purification data** (previously, only the hnRNP interaction was confirmed by Vierbuchen et al.).

Additionally, we performed **GRADR predictions for the Mail1 lncRNA**, which we previously identified as a cytoplasmic TLR-IRF3 signaling regulator interacting with the ubiquitin-reader OPTN (PMID: 32241891). While GRADR did not predict OPTN, the top-predicted cytoplasmic Mail1 interactors included SQSTM1, a known key interactor of OPTN in the ubiquitin-proteasome network (**Review Fig. 1C**). Among the top pathways predicted for GRADR-determined Mail1-interactors were **ubiquitin-proteasome related terms (Review Fig. 1D)**. This further supports the ability of GRADR to link lncRNAs to their relevant cellular machineries. These results on Mail1 are only included in Review Fig. 1 and not in the revised manuscript, as we are already using these data in another manuscript further dissecting the roles of the Mail1 lncRNA in the ubiquitin-proteasome system.

Overall, our benchmarking demonstrates that **GRADR effectively predicts RNA interactions with key protein machineries** (e.g., **Actin mRNA with ribosomal factors [Fig. 5]**, **RMRP RNA with mitochondrial factors [Fig. 5]**, **LUCAT1 with splicing-associated factors [Fig. S6]**, and **Mail1 with ubiquitin-proteasome factors [Review Fig. 1]**). While GRADR may not achieve the same specificity as targeted affinity purification approaches, it offers a global perspective on plausible interactions and helps refine candidate selection from noisy affinity purification data. We have **expanded the**

discussion section to further elaborate on the advantages and limitations of GRADR. Key revisions made in response to this reviewer comment include:

- **Results text revised** to describe the new data shown in Fig. S6 (page 7, line 13-20).
- **Methods revised** to include additional details on ChIRP and proteomics (page 13, line 43-44; page 16, lines 13-16; Table 4).
- **Dataset S9 expanded**, to include the new LUCAT1 ChIRP-MS data.
- **Discussion expanded** to elaborate on the advantages and limitations of GRADR (page 9, last paragraph).

Review Figure 1: Ability of GRADR to predict known lncRNA interactors. **A)** LUCAT1 ChIRP-MS result published by Vierbuchen et al. (screenshot from PNAS 2023; PMID: 36577072). **B)** Volcano plot showing lncRNAs detected in an SFPQ CLIP-seq experiment (fold-changes and p-values describe the detection in SFPQ-IgG compared to control-IgG based CLIPs; three independent experiments). Relevant lncRNAs are indicated (NEAT1 as a positive control for the experiment). **C)** GRADR-predicted MalL1 interacting proteins (SQSTM highlighted). **D)** ENRICHR based Bioplanet-pathway predictions for the MalL1 interactors determined by GRADR ($r \geq 0.5$, $p \leq 0.05$).

7. Figure 6A-C, by ChIRP-MS cross-validation, only a small fraction of ROCK1 interactors were recovered by GRADR (4 out of 21); the other 7 interactors were found by GRADR less further validation. I think it is better to perform a third experiment (e.g., ChIRP-WB), which would help readers evaluate the power of GRADR.

Response: In our response to point 6, we elaborate on the variability between GRADR and ChIRP-MS, as well as the inherent noise in RNA affinity purification experiments. This variability is evident even when ChIRP experiments for the same lncRNA are conducted by different labs, as demonstrated in Fig. S6E-G. Given these technical limitations, the relatively small overlap between GRADR and ChIRP-MS in our opinion reflects an advantage rather than a limitation: **GRADR may help refining noisy affinity purification results, filtering out false positives and focusing on a select number of plausible interactors, as discussed in our revised manuscript (page 9, last paragraph).**

To further assess the robustness of our GRADR predictions for ROCK1, we already benchmarked our results against machine-learning and eCLIP-based predictions, and performed cross-linking and immunoprecipitation (CLIP) experiments (Fig. S9A-C and Fig. 6A-D). However, we also considered the reviewer's suggestion and attempted additional validation using **ChIRP-WB with hnRNP detection**. As expected, due to the limited protein yield from ChIRP purifications and the higher material requirements of Western blotting compared to mass spectrometry, protein signals were only detectable in the input fraction but not in the eluates (**Review Fig. 2**). Due to the limited protein obtained in RNA-affinity purifications, we believe the **standard procedure to cross-validate ChIRP- or GRADR-based interactor predictions should be CLIP-seq or -qPCR (as conducted in Fig. 6D and in Review Fig. 1B).**

We believe that our multi-layered approach for ROCK1 (including GRADR, ChIRP-MS, CLIP-qPCR, machine-learning predictions, and eCLIP analysis) and our additional benchmarking of GRADR in the

revised manuscript (reply to point 6) provides strong validation of the predictive accuracy of GRADR within the recommended limits now explained in more detail in the discussion section (page 9, last paragraph).

Review Figure 2: ChIRP-Western-blot (ROCK1 capture and hnRNP detection). Control and ROCK1 ChIRP input and eluate samples were run on an SDS-PAGE gel, followed by Western blot and hnRNPL (left) and subsequent hnRNPA2B1 detection (right) (protein sizes and antibodies used are specified). Two independent experiments (n1 and n2) were conducted.

Minor points:

1. The SMyLR cannot be accessed without password.

Response: As documented on the last page of our manuscript the current **password for accessing SMyLR is: test123**. This ensures that reviewers can explore the database during the review process. The password-protection **will be lifted upon manuscript acceptance**, allowing unrestricted access.

2. Two small figures on the right side of Fig. 2B are poorly described.

Response: We thank the reviewer for bringing this to our attention. The missing description for these panels **has now been added** in the figure legend.

3. Names of lncRNA used in the manuscript should be consistent, e.g., MROCK1/ROCK1.

Response: We agree with the reviewer and unified the names of the lncRNAs throughout the manuscript.

4. Fig.3C and 3D, the authors reported that a series of mRNA/lncRNA are up/down regulated in the CRISPRi experiment. Did the authors use lncRNA over expression to validate this finding?

Response: We thank the reviewer for the suggestion. In this study, we did not use overexpression to validate the CRISPRi experiments. Instead, we used independent sgRNAs and performed RNA-seq and proteomics in triplicates to minimize off-target effects and false positives. Given the scope and workload of these comparative silencing experiments, adding overexpression-based datasets would have exceeded our capacities. However, we are already conducting **follow-up studies, focusing on LINC00158 and LINC01215, where we use lentiviral overexpression**. We previously validated this approach for PIRAT (Aznaourova et al., PNAS, 2022), where CRISPR and overexpression produced

opposing phenotypes, confirming the robustness of our systems. We hope the reviewer understands that further characterization of the five lncRNAs, including ectopic expression studies, would be beyond the scope of this manuscript and will be part of separate publications currently in preparation, detailing the roles and mechanisms of these lncRNAs in detail.

However, we did **further experiments to validate the function of the lncRNA ROCK1 in macrophages through another approach**. Specifically, we could show that **knockdown of ROCK1 in GM-CSF or M-CSF derived macrophages using RNAi** results in the **up-regulation of GATA2** expression (**new Fig. S9D**), similar to our CRISPR-based experiments with THP1 macrophages. Since ROCK1 is in the focus of the present manuscript, we included these additional data in the revised manuscript and updated the figure legend, **Results text (page 8, line 17-18)** and Methods (page 12, line 24-31, Table 3) accordingly. We believe, these results further support the robustness of our experimental procedures and of the results obtained.

Reviewer #3 (Remarks to the Author):

This study describes the development of GRADR which is a gradient based RNA interactor profiling tool. The goal is to develop a novel method allowing for the capture of global lncRNA protein interaction maps within macrophages. This is accompanied by a web user interface which makes this tool highly accessible especially to researchers that perhaps lack bioinformatic skills. Controls used are sound and mechanistic insights into ROCK1 as a confirmation of the tool in addition to the other lncRNA candidates is notable and provides confidence in the tool. There is a large amount of omics data available through this manuscript including RNA-seq data from various cell types, stimulations and proteomics. This data will be a welcome resource to the macrophage community and I only have a small number of suggestions and clarifications to make conclusions clearer.

Response: We thank the reviewer for acknowledging the utility of GRADR as an accessible tool for mapping lncRNA-protein interactions and are pleased that the reviewer views our omics datasets as a valuable resource for the macrophage research community. In response to the reviewer's suggestions, we have provided additional clarifications on the GRADR methodology, refined our discussion of its applications, and addressed all specific points raised to improve the clarity of our conclusions.

- Details on what GRADR actually does and the measures of success are very vaguely discussed in the manuscript. They mention that positive controls work which is great but providing additional numbers on how many false positives are found and what type of threshold should be set in order consider something a possible hit for follow up studies would be beneficial. Also after I took a quick look through the online web interface I looked up some of their example lncRNAs and all have a top hit of KHDRBS1, can the authors comment on why this protein which is presumably a false positive comes up at the top hit.

Response: We appreciate the reviewer's valuable feedback and have made several revisions to clarify the GRADR methodology, including thresholding criteria, and predicted interactomes:

To enhance transparency, we now provide a **detailed step-by-step protocol for GRADR** in the revised Supplementary Information, helping others to understand and reproduce the methodology. As stated in the Methods section, a Pearson correlation coefficient ($r \geq 0.5$) for co-sedimentation and an RNA-binding $p \leq 0.05$ are the thresholds we found useful for selecting potential interactors. These criteria have also been explicitly included in the step-by-step protocol (at the end of the Supplementary Information file).

Additionally, in response to Reviewer 2 (point 6), we **conducted new LUCAT1 ChIRP-MS experiments to benchmark GRADR against both published and our own targeted affinity purifications (Fig. S6D-G)**. Based on these new results, we have **expanded the discussion section** to address the overlap of GRADR predictions with ROCK1 and LUCAT1 interactomes identified by ChIRP-MS. We conclude that while GRADR can successfully predict interactors for follow-up studies, as exemplarily shown for several RNAs in our manuscript, its primary application should be to refine affinity purification results or predict larger cellular machineries or pathways within which a given lncRNA may function (**page 9, last paragraph**).

Regarding KHDRBS1, we re-examined our dataset, and **KHDRBS1 does not appear to be a false positive**. KHDRBS1 frequently emerges as a top interactor in RNA affinity purification experiments conducted in our lab. This is also reflected in the current manuscript, where **KHDRBS1 is identified in both GRADR predictions and ChIRP-MS data for LUCAT1 (Fig. S6D-E)**, and in **Fig. 6E and Fig. S8E**, where KHDRBS1 is detected as a **top interactor of the ROCK1-binding protein hnRNPL in our CLIP-MS**

experiment. Additionally, KHDRBS1 association with hnRNPs and splicing-related proteins is supported by external tools (e.g., R-DeeP: <https://rbp2go.dkfz.de/>) (Review Fig. 3A). Notably, KHDRBS1 is not the only protein frequently predicted as an interactor by GRADR. We believe this is due to the following reason: In our previously published GRAD-seq dataset (Aznaourova et al., PNAS 2020; Review Fig. 3B), we grouped lncRNAs into two categories:

- Group I: Non-ribosome-associated lncRNAs, likely noncoding
- Group II: Ribosome-co-sedimenting lncRNAs

Within group I, we identified a large, distinct subgroup (**lncRNA group Ia**) that shares **highly similar sedimentation profiles** and includes many LPS-responsive lncRNAs (Review Fig. 3B). Our ongoing research suggests that these lncRNAs are functionally linked to a common cellular machinery. While we cannot yet disclose the key RNA-binding protein (masked by “XXXX” in Review Fig. 3C) governing the sedimentation behavior of this large lncRNA cluster, **KHDRBS1 appears to be part of its interactome**. The existence of this large group Ia lncRNA subcluster, interacting with a common machinery (Review Fig. 3B-C, manuscript in prep.) likely explains why KHDRBS1 and other splicing related factors (e.g. hnRNPs) frequently appear in GRADR-predicted and experimentally derived lncRNA interactomes.

To improve data accessibility, we have now incorporated a **new feature in the SMYLR database allowing users to export all proteins from the GRADR plot into a text file**, including their Pearson r and p-values. This enables further filtering (e.g., Pearson r ≥ 0.5) and downstream pathway analyses.

We thank the reviewer for prompting these improvements, which we believe enhance the clarity and utility of GRADR for the broader research community.

Review Figure 3: **A)** Screenshot of KHDRBS1 interactome prediction by R-DeeP (<https://rbp2go.dkfz.de/>). **B)** 10-60 % glycerol gradient sedimentation profile (Grad-seq) for lncRNAs found to be expressed in LPS-stimulated G-M Φ (RPKM ≥ 0.5). Group Ia lncRNAs are highlighted (red rectangle). Horizontal axis: gradient fractions (10-60 %); vertical axis: lncRNAs; lncRNA abundance in each fraction illustrated through color-coded Row-Z-scores. **C)** Grad-seq profile of RNAs recovered in CLIP-seq experiments with a fold-change ≥ 10 or ≤ 2 based on three independent CLIP-experiments with a target-protein (XXXX) specific IgG compared to a control IgG. Significant differences in the abundance of RNAs in each fraction (comparing these two groups) are denoted by asterisks (ANOVA test; p ≤ 0.05). Grey horizontal bar indicates the area where the XXXX-protein is most abundant on the gradient (exceeding a base-mean fold-change of 1, as determined by Grad-seq mass-spectrometry).

• CHIRP on ROCK1, appears to be only a single replicate comparing direct CHIRP to control from GSE268546. Can the authors comments on this

Response: The reviewer is correct that we presented ChIRP-seq results from a single sequencing experiment. Unfortunately, during manuscript preparation, our service provider for ChIRP-seq library

preparation and sequencing (Vertis Biotech AG) ceased operations, and we are currently in the process of identifying a suitable alternative, offering CLIP-seq results of the same quality.

However, **we performed the ChIRP experiment in triplicates and validated the results using qPCR quantifications**. These confirm significant enrichment of the ROCK1 locus, while the MARCKS-upstream region was not enriched (validated with two independent primer sets). We apologize for not including these data in the initial manuscript. They have now been integrated into **Fig. 6F**. The Methods (page 13, line 37-40) and Fig. 6 legend have been updated accordingly.

- GSE269547, the token provided to the reviewer did not work and therefore I was unable to evaluate this data

Response: We would like to inform the reviewer that access to the NGS data deposited in NCBI GEO under GSE268547 (accessible at <https://www.ncbi.nlm.nih.gov/geo/query/acc.cgi?acc=GSE268547> with the reviewer token `ufknsoajdgrvud`) is functioning correctly on our end.

Could there be a possibility that the reviewer referred to the wrong GSE number? The correct dataset ID is GSE268547, whereas the reviewer mentioned GSE269547. Additionally, we would like to note that the two NCBI GEO datasets we uploaded (GSE268546 and GSE268547) have different reviewer tokens, both of which are provided at the end of our manuscript.

- Table S1: For all CRISPRi experiments it is unclear what the control is in each case. Are they using multiple nontargeting controls for the experiments? This is also not clear in the figure legends for Figure 3 and 4

Response: We recognize the need for additional clarification regarding the controls used in our CRISPRi experiments. Rather than using non-targeting guide RNAs, which carry the risk of unintended gene targeting and could introduce confounding effects, we opted to use a uniform control: cells transduced with the same CRISPRi vector but lacking a guide RNA insert. This approach ensures that all lncRNA knockdowns are comparable (11 knockdowns were generated for this experiment; Fig. S4A). Additionally, since our focus was on the differences between individual lncRNA knockdowns, each lncRNA silencing in Fig. 3-4 effectively serves as an additional internal reference for the others.

This strategy is supported by our data: in the Cytoscape network plot (Fig. 3D), no significant mRNA regulation event is observed consistently across all five lncRNA knockdown datasets, indicating the absence of systematic confounders. Furthermore, in the revised manuscript, for ROCK1, we additionally validated the key mRNA regulation event identified (GATA2 upregulation) using an alternative silencing approach (RNAi; Fig. S9D), reinforcing the reliability of our results.

To clarify this in the manuscript, we have **revised the relevant Methods paragraph** to explicitly state: "Control cell lines were generated by transducing cells with the same empty CRISPRi vector, lacking a guide RNA insert" (**page 12, line 21-22**). Additionally, we have updated the Fig. 3C legend and NCBI GEO GSE268547 entry for improved clarity.

Minor point

- Fig 6C. A sized venn diagram is preferable

Response: We agree with the reviewer and have **changed the Venn diagram in Fig. 6C accordingly**.

Further changes introduced during the revision process:

- We identified a filtering error in the heatmaps presented in Figs. 2B and S3F, where some lncRNAs that did not meet the ≥ 2 -fold regulation threshold were inadvertently included, while others that met the threshold were omitted. This threshold was consistently applied throughout our manuscript for identifying immune-responsive lncRNAs. The issue has now been corrected, and the updated figures accurately display all lncRNAs upregulated ≥ 2 -fold in aM Φ and G-M Φ , identified in Fig 1E.
- In our original manuscript we stated that THP1 cells were stimulated with LPS for 4 h. However, the correct stimulation duration was 8 h. Based on our experience, 8-hour stimulation in THP1 cells yields a response comparable to 4-hour stimulation in primary macrophages (the LPS-response of THP1 cells is delayed). This has now been corrected, ensuring that all stimulation time points are accurately reported.
- Minor spelling mistakes were corrected throughout the manuscript and the terms G-M Φ and M-M Φ as well as the spelling of all lncRNAs reported were unified.
- Some article sections were reordered and terms such as novel and unprecedented were replaced or removed, to meet Nature Communications formatting specifications.
- The last paragraph of the introduction section was modified to meet Nature Communications specifications (e.g. use of present tense).
- In the SMyLR database, the X-axis labelling for the GRADR plot was corrected (“Pearson r ” instead of “ R^2 ”) and the heading was changed to “GRADR-Predicted Interactors (top 100; mouse-over)”, to specify according to which criterium the interactors shown were selected.
- In the SMyLR database legend, the axis for the radar plot was specified.